# Beyond Uniformity: Regularizing Implicit Neural Representations through a Lipschitz Lens

**Julian McGinnis**[1,2,⋆]    **Suprosanna Shit**[3,4,⋆]    **Florian A. Hölzl**[1,5]    **Paul Friedrich**[6]
**Paul Büschl**[3]    **Vasiliki Sideri-Lampretsa**[1]    **Mark Mühlau**[1]    **Philippe C. Cattin**[6]
**Bjoern Menze**[3,4]    **Daniel Rueckert**[1,2,7,†]    **Benedikt Wiestler**[1,2,†]

[1]Technical University of Munich    [2]Munich Center for Machine Learning    [3]University of Zurich
[4]ETH AI Centre    [5]Hasso-Plattner-Institute    [6]University of Basel    [7]Imperial College London

[⋆]Equal contribution    [†]Equal senior contribution

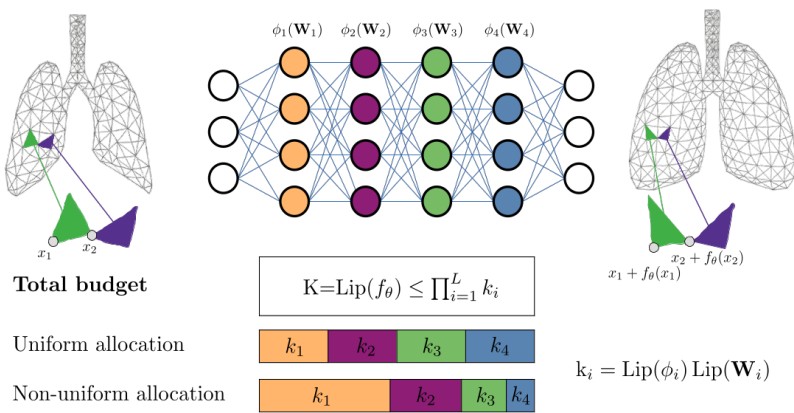

Figure 1: **Bridging the gap between theory and practice in Lipschitz regularization.** While Lipschitz continuity offers a principled form of implicit regularization, *selecting* and *distributing* the budget $K$ remains non-trivial. We propose a novel, data-driven approach to derive $K$ from interpretable, signal- or domain-specific properties, such as tissue compressibility in deformable registration, and strategically distribute this budget across layers. This methodology allows for a better balance of smoothness and expressiveness compared to uniform allocation strategies.

## Abstract

Implicit Neural Representations (INRs) have shown great promise in solving inverse problems, but their lack of inherent regularization often leads to a trade-off between expressiveness and smoothness. While Lipschitz continuity presents a principled form of implicit regularization, it is often applied as a rigid, uniform 1-Lipschitz constraint, limiting its potential in inverse problems. In this work, we reframe Lipschitz regularization as a flexible *Lipschitz budget framework*. We propose a method to first derive a principled, task-specific total budget $K$, then proceed to distribute this budget *non-uniformly* across all network components, including linear weights, activations, and embeddings. Across extensive experiments on deformable registration and image inpainting, we show that non-uniform allocation strategies provide a measure to balance regularization and expressiveness within the specified global budget. Our *Lipschitz lens* introduces an alternative, interpretable perspective to Neural Tangent Kernel (NTK) and Fourier analysis frameworks in INRs, offering practitioners actionable principles for improving network architecture and performance. Code and experimental results are available at: https://lipschitz-inrs.github.io.

# 1 INTRODUCTION

Implicit neural representations (Tancik et al., 2020; Sitzmann et al., 2020) have emerged as a promising modality-agnostic modeling framework, with applications ranging from data compression (Strümpler et al., 2022; Dupont et al., 2022b; Wu et al., 2025), representation learning (Dupont et al., 2022a; Bauer et al., 2023; Friedrich et al., 2025), novel view synthesis (Mildenhall et al., 2021; Barron et al., 2021), to inverse problems including deformable registration (Wolterink et al., 2022; Sideri-Lampretsa et al., 2024), MRI reconstruction (Corona-Figueroa et al., 2022; McGinnis et al., 2023; Huang et al., 2023), and population modeling (Bieder et al., 2024; Dannecker et al., 2024).

Despite their widespread success, a significant gap remains in our understanding of the generalization mechanisms underlying INRs. Notably, studies by (Fridovich-Keil et al., 2022) and (Kim & Fridovich-Keil, 2025) reveal that regularized grids can match or even outperform non-regularized INRs, challenging the presumed superiority of INRs in computer vision. These findings may be intuitively explained by the absence of implicit regularization in expressive INR architectures (Ramasinghe et al., 2022) and may be addressed by incorporating explicit regularization into the learning process (Gropp et al., 2020; Niemeyer et al., 2022). Alternatively, one can employ spectrally constrained architectures (Liu et al., 2022; Coiffier & Béthune, 2024; Zhang et al., 2023) as implicit regularization, which we systematically study in this paper.

In these applications, the network is typically constrained to be 1-Lipschitz, which we refer to as a *Lipschitz Budget* of one. This choice of unit Lipschitz constant is theoretically well-founded: the Wasserstein distance can be reformulated as a maximization over 1-Lipschitz functions through the Kantorovich-Rubinstein duality (Arjovsky et al., 2017), while for SDFs, the constraint naturally emerges from the triangle inequality and Eikonal equation (Coiffier & Béthune, 2024). To globally fulfill the *Lipschitz Budget* of one, current methods typically set all network component bounds to 1-Lipschitz, thus *allocating* the total *Lipschitz Budget* [1] uniformly across all network components, including layers, activations, and embeddings. *However, we question whether this uniform distribution of the budget is optimal.*

In this work, we propose a new perspective on the regularization of INRs by analyzing it through the lens of Lipschitz continuity. We advocate for shifting the paradigm from a rigid, uniform Lipschitz constraint to a more flexible, budgeted strategy. This approach directly addresses two fundamental, open problems in the Lipschitz regularization literature (Gouk et al., 2021; Newhouse et al., 2025): (1) how to determine a principled total Lipschitz budget $K$ for a given task, and (2) how this budget should be optimally allocated among the network's components to maximize performance. Our primary contributions are as follows:

1. We derive induced Lipschitz constants for INR components providing a framework to move beyond rigid 1-Lipschitz constraints. This allows for task-specific regularization strength.

2. We propose to estimate the Lipschitz budget guided by domain knowledge (deformable image registration) or via an interpretable oracle based on expected signal variation within the target domain (inpainting), and set it as an upper Lipschitz bound of the network.

3. Proposing a non-uniform budget allocation strategy, distributing the $K$-Lipschitz budget across network components based on their intrinsic Lipschitz characteristics, allows for balancing expressiveness and smoothness within the network.

4. Finally, we show how viewing empirically effective strategies, such as weight scaling, through a Lipschitz lens can provide an important, complementary perspective to Fourier analysis and the Neural Tangent Kernel theory in explaining their effectiveness.

# 2 BACKGROUND AND RELATED WORK

**INRs and Regularization**: Implicit representations have seen increasing adoption since neural radiance fields (Mildenhall et al., 2021), with coordinate encodings (Rahimi & Recht, 2007; Tancik et al., 2020; Müller et al., 2022), activation functions (Sitzmann et al., 2020; Ramasinghe & Lucey, 2022; Saragadam et al., 2023), and optimization (McGinnis et al., 2025) being crucial for network expressivity to enable detailed modeling. Recent architectural works target the inherent trade-off

---

[1]Total Lipschitz of a neural network is the product of each component's Lipschitz, defined in subsection 3.1.

of high expressiveness and smoothness (Chen et al., 2023; Kazerouni et al., 2024; Liu et al., 2024; Yeom et al., 2024), but so far, no principled approach to network design exists. To balance expressiveness and generalization, explicit regularization has been extensively studied (Niemeyer et al., 2022; Yang et al., 2023; Wynn & Turmukhambetov, 2023), especially in shape modeling (Gropp et al., 2020; Atzmon & Lipman, 2020; Ben-Shabat et al., 2022). While explicit regularization can be beneficial, it only rigidly constrains model capacity and does not inherently limit the network's susceptibility to input perturbations. We deem this particularly relevant in the case of INRs, where early works have shown implicit regularization to be largely absent (Ramasinghe & Lucey, 2022; Ramasinghe et al., 2022), putting this at the core of our study.

**Lipschitz Constraints:** The use of Lipschitz constraints has been widely studied in both theory (Szegedy et al., 2013; Bartlett et al., 2017; Gouk et al., 2021) and application (Arjovsky et al., 2017; Gulrajani et al., 2017). However, within neural fields, it remains relatively underexplored. We believe systematically studying this for INRs is timely, especially as spectral regularization has recently gained traction in computer vision (Kim et al., 2021; Qi et al., 2023) and large language models (Kim et al., 2021; Large et al., 2024; Newhouse et al., 2025). While Liu et al. (2022) regularizes the upper Lipschitz bound in conditional shape modeling to enable meaningful inter- and extrapolation, most other works are limited to 1-Lipschitz networks. Coiffier & Béthune (2024) have introduced 1-Lipschitz neural distance fields, Mujkanovic et al. (2024) for neural Gaussian scale-space fields, and Zhang et al. (2023) for vision applications. However, solely relying on uniform allocation of 1-Lipschitz layers, without task-specific budgets or non-uniform allocations, severely limits a network's expressivity.

To enable more expressive Lipschitz bounds requires enforcing constraints in the INRs' linear layers. Most commonly, this is achieved with spectral normalization (Golub & Van Loan, 2013), or efficient approximations thereof using power iteration methods (Miyato et al., 2018). For stricter constraints, Björck orthonormalization (Björck & Bowie, 1971; Anil et al., 2019) iteratively refines matrix orthogonality, which inherently maintains unit spectral norm. Alternatively, directly developing Lipschitz constrained layers is a topic of active study (Serrurier et al., 2020; Prach & Lampert, 2022; Prach et al., 2024; Araujo et al., 2023).

*Gradient norm–preserving activations* (Anil et al., 2019; Singla et al., 2021; Ducotterd et al., 2024) are designed to approach the upper bound of a network's Lipschitz capacity, in contrast to standard activations like ReLU. Complementary methods aim to approximate the actual Lipschitz constant, both for ReLU-based MLPs (Virmaux & Scaman, 2018; Fathony et al., 2020) and, more recently, for networks with gradient-preserving activations (Pauli et al., 2024a;b). This work builds upon previous work by providing a unified framework for INR components. We investigate the critical questions of deriving task-specific Lipschitz budgets and non-uniform allocation for INRs to tackle the expressiveness-smoothness trade-off. In the following, we provide the theoretical background on Lipschitz theory for this work.

## 3 LIPSCHITZ COMPOSITION OF INRS

### 3.1 LAYER-WISE LIPSCHITZ CONSTANTS

A function $f : \mathbb{R}^d \to \mathbb{R}^m$ is Lipschitz-continuous with constant $K$ if $\|f(\boldsymbol{x}_1) - f(\boldsymbol{x}_2)\| \leq K\|\boldsymbol{x}_1 - \boldsymbol{x}_2\|$ for all $\boldsymbol{x}_1, \boldsymbol{x}_2 \in \mathbb{R}^d$. This Lipschitz constant $K$ bounds the maximum rate of change of the function. In the context of INRs, we argue that controlling the Lipschitz constant is crucial for balancing smoothness and expressiveness. For a network $f_\theta$ with $L$ layers, the composition property of Lipschitz continuity dictates that the overall Lipschitz constant $K$ is bounded above by the product of the Lipschitz constants of individual layers and activations $\phi$:

$$K = \text{Lip}(f_\theta) \leq \prod_{i=1}^{L} \text{Lip}(\phi_i) \, \text{Lip}(\boldsymbol{W}_i) \, , \tag{1}$$

where $\boldsymbol{W}_i$ represents the weight matrix of the $i$-th linear layer and $\phi_i$ is the activation function applied after it. This compositional property represents the cornerstone of our framework. It reveals that a *global budget K* can be achieved through a vast combination of layer-wise constants, which motivates our central exploration of different budget allocation strategies.

Linear layers are the fundamental building blocks of INRs. The Lipschitz constant of a linear layer is determined by its weight matrix $\boldsymbol{W}_i$, specifically through the induced operator norm. The spectral norm, defined as the largest singular value of $\boldsymbol{W}_i$, is a common choice: $\text{Lip}(\boldsymbol{W}_i) = \|\boldsymbol{W}_i\|_2 = \sigma_{max}(\boldsymbol{W}_i)$. Constraining the norm of a linear layer is achieved through techniques like spectral normalization, *e.g.* using power iteration (Miyato et al., 2018) to estimate the singular value decomposition, and orthogonalization (Björck & Bowie, 1971; Anil et al., 2019), which forces the weight matrix to be orthogonal, such that $\boldsymbol{W}_i^\top \boldsymbol{W}_i = \boldsymbol{I}$.

For some INRs, the first layer is a feature projection $\gamma(\cdot)$ with an associated Lipschitz constant. For positional encodings (Mildenhall et al., 2021), given a mapping $\gamma_{\boldsymbol{p}}(\boldsymbol{p}) = (\sin(2^0\pi\boldsymbol{p}), \cos(2^0\pi\boldsymbol{p}), \ldots, \sin(2^{L-1}\pi\boldsymbol{p}), \cos(2^{L-1}\pi\boldsymbol{p}))$ of 1D input $\boldsymbol{p}$ to a vector of sinusoidal functions, the Lipschitz constant is $\text{Lip}(\gamma_{\boldsymbol{p}}) = \pi\sqrt{\frac{4^L-1}{3}}$. Random Fourier Features $\gamma_f(\boldsymbol{v})$ (Rahimi & Recht, 2007; Tancik et al., 2020) similarly map inputs using random projections with a Lipschitz constant of $\text{Lip}(\gamma_f) = 2\pi\sqrt{\lambda_{\max}\left(\sum_{j=1}^m b_j b_j^\top\right)}$. We refer to Appendix E for the exact derivation.

Lastly, we analyze activation functions as a crucial non-linear building block of INRs. Activations can be broadly categorized into two groups. 1-Lipschitz activations, like ReLU ($\phi(x) = \max(0, x)$) and GroupSort (Anil et al., 2019), as well as related MaxMin and FullSort, inherently bound the change in output relative to the input, simplifying Lipschitz control. Learnable activation functions can be constrained to maintain a Lipschitz constant of 1 through parameter regularization. Non-1-Lipschitz activations usually have hyperparameter-dependent Lipschitz bounds, such as the frequency $\omega$ or the scale parameter $a$. For sinusoidal activations, we have $\phi(x) = \sin(\omega x)$ with derivative $\phi'(x) = \omega\cos(\omega x)$. Since $\max_x |\cos(\omega x)| = 1$, we obtain $\text{Lip}(\phi_{\text{Sinusoidal}}) = \omega$. For Gaussian activations with $\phi(x) = e^{-\frac{x^2}{2a^2}}$, the derivative is $\phi'(x) = -\frac{x}{a^2}e^{-\frac{x^2}{2a^2}}$. To find the maximum of $|\phi'(x)|$, we solve $\frac{d}{dx}|\phi'(x)| = 0$, which occurs at $x = \pm a$. Evaluating at $|\phi'(\pm a)| = \frac{1}{a}e^{-\frac{1}{2}}$ yields $\text{Lip}(\phi_{\text{Gaussian}}) = \frac{1}{a\sqrt{e}}$. Non-1-Lipschitz activations typically require careful scaling or regularization to prevent unbounded growth in the overall Lipschitz constant. The choice of activation function and its parameterization significantly impacts network behavior.

## 3.2 Lipschitz Budget Allocation

Given the total Lipschitz budget $K_B$, we need to allocate this to individual network components. Let's consider that we have a total of $M$ network components. For this, Eq. 1 is used for computing the total Lipschitz of the network. The allocation problem is to find $[K_i]_{i=1}^M$ which satisfy $\prod_{i=1}^M K_i = K_B$ where individual $K_i > 0$. This product allocation can be thought of as a total sum allocation in logspace. Besides uniform allocation (A), we also study four non-uniform allocation strategies (B-E):

**(A) Uniform allocation.** In this work, we consider a baseline uniform allocation strategy, where we allocate each component the same Lipschitz contribution. This results in $K_i = \sqrt[M]{K_B}$.

**(B) All-first allocation.** We allocate $K_1 = K_B$ and $K_i = 1$ for $i = 2 : M$. This is motivated by the intuition that the first layer often plays a critical role in feature learning for neural networks, where early weights have been shown to critically influence network sensitivity (Raghu et al., 2017).

Additionally, we consider monotonically decreasing parametric allocation using linear, exponential, and cosine-annealed strategies constrained by the minimal Lipschitz of the last layer $K_M = K_{min}$. Let $t_i = \frac{i-1}{M-1}$ for $i = 1 : M$ with $t_1 = 0$ and $t_M = 1$ and define $u_i = \log K_i$.

**(C) Linear allocation.** We impose a straight line in *budget* space from an unknown $K_1 = s_0$ to $K_M = K_{\min} : K_i = s_0 + (K_{\min} - s_0)\, t_i, \quad i = 1 : M$, and choose $s_0 > 0$ as the unique solution of $\sum_{i=1}^M \log\big(s_0 + (K_{\min} - s_0)\, t_i\big) = \log K_B$. using a one-dimensional root finder such as the bisection/Newton method.

**(D) Exponential allocation.** We use a front-heavy ramp in *log* space as follows

$$u_i = \log K_{\min} + \frac{\log K_B - M \log K_{\min}}{\sum_{j=1}^M (1 - t_j)}(1 - t_i), \qquad K_i = \exp(u_i).$$

**(E) Cosine-annealed allocation.** We consider the following $g(t_i) = \frac{1+\cos(\pi t_i)}{2}$, so $g(t_1) = 1, g(t_M) = 0$. We seek a single amplitude $\alpha > -1$ such that

$$K_i = K_{\min}\left(1 + \alpha\,g(t_i)\right) \text{ for } i = 1 : M, \text{ subject to } K_{\min}^M \prod_{i=1}^{M-1}\left(1 + \alpha\,g(t_i)\right) = K_B.$$

Equivalently, $\alpha$ is the unique solution to the 1-D monotone equation $\sum_{i=1}^{M-1}\log\left(1 + \alpha\,g(t_i)\right) = \log\left(\frac{K_B}{K_{\min}^M}\right), \alpha > -1$. Exemplary allocation strategies are provided in the supplementary Fig. 9.

## 4 REVISITING 1-LIPSCHITZ UNDER FLEXIBLE BUDGET ALLOCATION

### 4.1 1-LIPSCHITZ SIGNED DISTANCE FIELDS

**Definition:** Let us consider an SDF as a learnable, implicit neural function $f_\theta$ which maps each coordinate of an input space $\Omega$ to its shortest orthogonal distance from that point to the boundary, i.e., to the surface of a geometric object. The sign of the function's output encodes a spatial relationship: negative values indicate that the coordinate resides inside the represented object, while positive values indicate that the coordinates are outside the object's boundary. We can thus express the shape's boundary as the function's zero-level set (Park et al., 2019; Coiffier & Béthune, 2024).

**1-Lipschitz:** Since SDFs should represent the *shortest, orthogonal* distance to a shape's boundary, the gradient magnitude is required to be $\|\nabla f\| = 1$ almost everywhere (Coiffier & Béthune, 2024). This property is known as the eikonal equation and holds for a *true* SDF throughout its domain, except at points where it is not differentiable, i.e., at the non-smooth boundary points. Adhering to this property ensures the function is 1-Lipschitz, which is critical for algorithms like ray marching (Hart, 1996) and further also enables the use of specific losses such as the hinge-Kantorovitch-Rubinstein loss for training neural distance fields (Coiffier & Béthune, 2024).

### 4.2 ON CAPACITY IN 1-LIPSCHITZ SDFS

1-Lipschitz SDFs can be achieved through various architectures and components; therefore, our first objective is to systematically evaluate the impact of two key design choices: (1) different spectral normalization techniques applied to linear layers, and (2) the incorporation of 1-Lipschitz activation functions, which, to the best of our knowledge, have not been explored in the neural SDF literature. Following the approach of Anil et al. (2019) for Wasserstein GANs, we investigate how achieving tighter Lipschitz bounds influences the learned SDF.

**Experiment 1 - Lipschitz Capacity:** Given the widespread use of ReLU activation functions in implicit neural representations for SDFs (Park et al., 2019; Gropp et al., 2020; Davies et al., 2020; Liu et al., 2022; Coiffier & Béthune, 2024), we investigate how imposing spectral normalization as a 1-Lipschitz constraint affects SDF learning quality. We evaluate this approach by training neural representations of the *Stanford bunny* model, comparing reconstruction quality against non-regularized SDFs using Chamfer Distance. To address the potential limitations of ReLU activations under Lipschitz constraints, we further examine how gradient-norm preserving activation functions can improve representational capacity by evaluating common gradient-preserving functions, including MaxMin (Anil et al., 2019), FullSort (Anil et al., 2019), and Householder activations (Singla et al., 2021) as implemented in Serrurier et al. (2020) against 1-Lipschitz ReLU-based SDF representations. To substantiate our claims, we also measure the network's Lipschitz constant using the empirical method presented in (Wang & Manchester, 2023b).

Our results in Figure 2 demonstrate that Björck orthonormalization (Björck & Bowie, 1971; Anil et al., 2019) and SLL (Araujo et al., 2023) achieve sharper reconstruction quality compared to standard spectral normalization (Miyato et al., 2018). Similarly, gradient-preserving activation functions, particularly MaxMin and Householder, also enable more effective utilization of the network's representational budget, allowing for finer geometric detail preservation. Notably, we find that the concept of Lipschitz capacity usage with respect to its empirically measured Lipschitz constant, proves intuitively interpretable for SDFs, correlating strongly with perceptual quality metrics: The closer a network is to utilizing the entire 1-Lipschitz budget, the better the image quality. Remarkably, with suitable spectral regularization, we observe that certain 1-Lipschitz activations provide

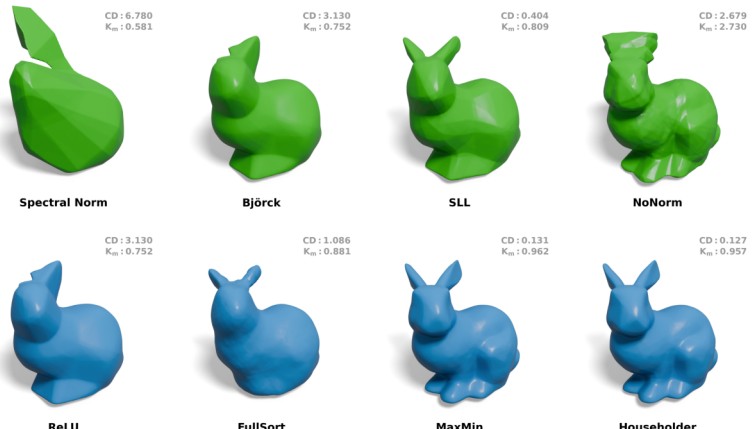

Figure 2: Learning the classical *Stanford bunny* shape with different spectral normalization techniques (first row) and different gradient-preserving activation functions in combination with a Björck normalized linear layer (second row) demonstrates that approaching the upper Lipschitz bound in SDFs correlates with perceptual quality. We report the Chamfer Distance ($\downarrow$) and the empirically estimated Lipschitz constant $K_m$ employing the estimation strategy in Wang & Manchester (2023a)for all reconstructions. Please refer to the Appendix for experimental details and results on other shapes. All shape visualizations are rendered using the excellent tools provided in Liu (2018).

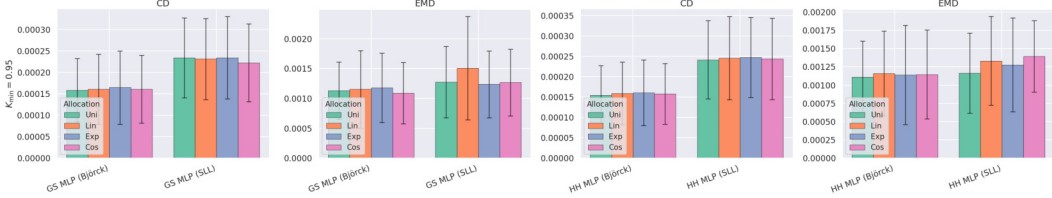

Figure 3: Budget allocation experiments for 1-Lipschitz SDFs. We train ten SDFs with Householder (HH) and GroupSort (MaxMin) activations, using Björck and SLL normalization. Non-uniform allocation offers negligible advantages over uniform constraints, confirming that standard strategies suffice for shape representation. Experimental details are described in subsection D.1.

strong alternatives to the widely established ReLU activation in shapes (Park et al., 2019; Davies et al., 2020; Coiffier & Béthune, 2024).

**Experiment 2 - Budget Distribution:** Up to this point, we have applied uniform 1-Lipschitz constraints across all layers of the neural network. However, the potential benefits of non-uniform constraint allocation remain largely unexplored (Gouk et al., 2021). This gap is particularly intriguing given the theoretical insights of Raghu et al. (2017), who demonstrate that neural network behavior exhibits high sensitivity to initial weight distributions, suggesting that layer-wise constraint variation could yield improved reconstruction capabilities. Motivated by this theoretical foundation, we investigate a strategic non-uniform allocation approach: we relax Lipschitz constraints in early layers to enable more expressive feature transformations, while imposing progressively tighter constraints in later layers. This allocation strategy maintains the global 1-Lipschitz budget through the composition property of Lipschitz functions outlined in subsection 3.1, ensuring theoretical guarantees while potentially enhancing representational capacity. The results presented in Figure 3 show that non-uniform allocation strategies perform on par with the standard uniform approach.

This result suggests that the benefits of non-uniform allocation may be stifled by the overly restrictive nature of a unit budget. This motivates our extension to the general K-Lipschitz setting, where the budget itself is a meaningful, task-dependent parameter.

## 5 BUDGET ESTIMATES & ALLOCATION STRATEGIES IN $K$-LIPSCHITZ INRS

In this section, we extend our investigation to a broader class of $K$-Lipschitz INRs for inverse problems, with an exemplary study of budget estimation and allocation within the context of deformable image registration for medical imaging using a knowledge-driven estimate, and an oracle-driven estimate for the classical computer vision application of inpainting. We include a third experiment for single-image super-resolution in Appendix B.

### 5.1 DOMAIN-DRIVEN BUDGETS IN THE $K$-LIPSCHITZ SETTING

**INRs within the registration landscape:** While deep learning models have been proposed to accelerate medical image registration from pairwise to cohort-level settings (Balakrishnan et al., 2018; 2019; Dalca et al., 2019), conventional optimization-based methods (Rueckert et al., 2002; Avants et al., 2011; Yushkevich et al., 2016) remain the gold standard for pairwise registration. Recently, INRs have emerged as a promising learning-based alternative, showing strong potential in lung CT and brain MRI registration (Wolterink et al., 2022; Sideri-Lampretsa et al., 2024). However, due to the lack of implicit regularization (Ramasinghe et al., 2022), explicit constraints such as bending energy (Rueckert et al., 2002), elastic (Burger et al., 2013), and curvature-based regularizers (Fischer & Modersitzki, 2003) are often required. Even then, implicit deformation fields tend to fold (Sideri-Lampretsa et al., 2024), motivating our investigation of spectral regularization as a form of implicit regularization within our $K$-Lipschitz framework.

**Lipschitz budget in K-Lipschitz displacement fields:** Accurate deformable registration of lung images is essential for monitoring COPD (Murphy et al., 2012; Galbán et al., 2012; Castillo et al., 2013), but respiratory motion and disease-related changes make this task challenging (Wolterink et al., 2022). While lung tissue is deformable, it preserves structural integrity and cannot undergo extreme distortions. Enforcing a Lipschitz constraint provides a principled upper bound on local stretching or compression, ensuring anatomically plausible displacement fields and preventing folding, tearing, or excessive compression. Clinical evidence suggests that a strain approaching 2.0 marks a threshold for tissue failure (Chiumello et al., 2008; Brower et al., 2008; Fung, 2013), motivating our choice of $K_B = 2$. Guided by this principle, we study allocation strategies on a lung dataset (Castillo et al., 2013) using SIREN (Wolterink et al., 2022; Sideri-Lampretsa et al., 2024) and ReLU FFN, both shown to be promising for modeling continuous displacement field, and evaluate performance via target registration error (TRE) and folding ratio of deformation field.

**Results:** While we observed no significant differences between allocation strategies in SDFs, we discovered important insights regarding training dynamics in deformable registration tasks that lead to concrete recommendations for practitioners. Our systematic evaluation reveals that the combination of network architecture and Lipschitz normalization method significantly impacts training stability and registration quality. We found that FFN using Björk or SLL normalization (Araujo et al., 2023), as well as SIREN with standard spectral normalization (Miyato et al., 2018), can exhibit training instabilities when applied to registration tasks. Based on these findings, we recommend two stable and effective config-

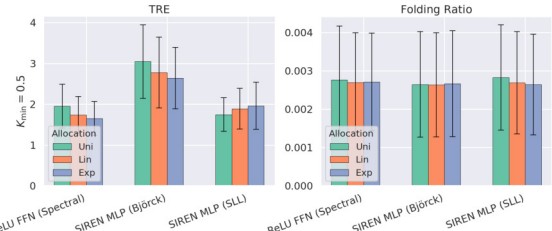

Figure 4: Comparison of smoothness (Folding Ratio ($\downarrow$)) and expressiveness (TRE ($\downarrow$)) across three Lipschitz-regularized INR architectures and allocation strategies. Non-uniform allocations (e.g., exponential) can improve TRE while maintaining a comparable folding ratio to uniform allocation.

urations: spectral ReLU FFN and SIREN networks with Björk or SLL normalization (c.f. Fig 4). These architectures consistently achieve stable training while maintaining the intended benefits of Lipschitz regularization, delivering a balance between TRE and folding ratio. Furthermore, we observe that the choice of allocation strategy (c.f. Fig 4 provides control over this trade-off: non-uniform strategies enhance model expressiveness and reduce TRE, while maintaining a comparable folding ratio to uniform strategies. These insights may enable practitioners to select configurations that best match their specific accuracy and smoothness requirements.

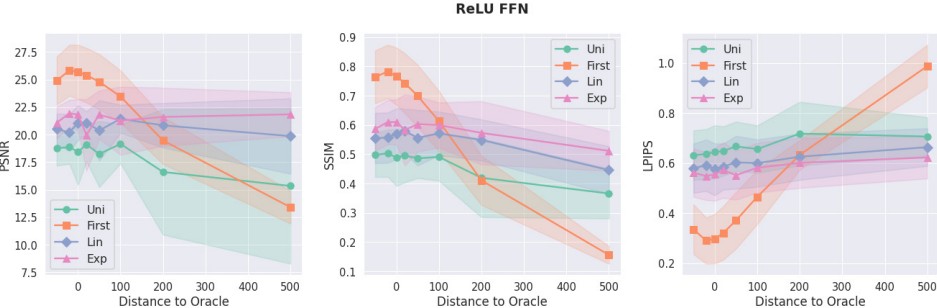

Figure 5: Results for different allocation strategies in the inpainting experiment using FFNs with $K_{min} = 1.0$ and SLL-normalization layers.

## 5.2 Data-driven Budget for $K$-Lipschitz Settings

A well-studied application of INRs in computer vision is image inpainting (Pathak et al., 2016; Yu et al., 2018), which aims to fill in missing or corrupted regions of an image. INRs are well-suited for this task (Sitzmann et al., 2020) as they represent images as continuous functions that can be queried at any spatial coordinate, enabling natural interpolation of missing pixel values.

**Definition:** Let's consider image inpainting as the task of reconstructing a complete image $\hat{I}$ from a corrupted image $I$ with missing or masked regions $M$. The goal is to fill in the missing pixels with plausible content. Given an input coordinate $(x, y) \in \Omega$, where $\Omega$ is the 2D image domain, an INR $f_\theta$ learns to map each spatial coordinate to its corresponding RGB color value: $f_\theta : (x, y) \to (R, G, B)$ For coordinates in the known regions, i.e., $(x, y) \notin M$, the network is trained to reproduce the original pixel values: $f_\theta(x, y) \approx I(x, y)$ for $(x, y) \notin M$ For masked coordinates, i.e., $(x, y) \in M$, the network infers appropriate color values based on the learned continuous representation, effectively completing the image by querying $f_\theta$ at the missing coordinate locations. The reconstructed image is then given by: $\hat{I}(x, y) = f_\theta(x, y)$ for all $(x, y) \in \Omega$

**A Lipschitz oracle for INRs:** We begin by discussing two useful oracles in INRs. For image reconstruction, i.e., inpainting experiments, we assume the image to have a bandwidth B, and assume that Lipschitz signal observation may follow this expectation. In this case, we can then state that (1) the Lipschitz constant is upper-bounded by the sampling distribution over a bounded input and output domain. (2) We decide to use an L2-norm-based approach (since we are spectrally constraining with L2), to estimate a lower-bounded Lipschitz constant using a gradient-based estimate. We refer to Appendix C for the oracle estimate.

**Experiment:** For the inpainting experiments, we use the CelebA dataset (Liu et al., 2015) and train several common INR architectures, including SIREN (Sitzmann et al., 2020), FFNs (Tancik et al., 2020), and Gaussian-activated INRs (Ramasinghe & Lucey, 2022), using the allocation strategies presented earlier. We evaluate performance using standard metrics: PSNR ($\uparrow$), SSIM ($\uparrow$) (Wang et al., 2004), and LPIPS ($\downarrow$) (Zhang et al., 2018). We present exemplary quantitative (Fig. 5) and qualitative (Fig. 6) results and refer to Appendix D for setup details, G.4 for statistical significance tests and G.5 for additional results.

As shown in Fig.6, non-uniform budget allocation strategies yield statistically significant improvements in reconstruction (c.f. G.4), peaking near the oracle estimate. Conversely, as shown in Fig. 5, performance degrades when the Lipschitz budget deviates from the oracle, particularly when using higher computational budgets (shown here for FFNs). This demonstrates that the oracle provides a meaningful approximation of the upper Lipschitz bound for the inpainting task. Moreover, as shown in G.5, different INR architectures exhibit varying degrees of self-regulation when deviating from the estimated oracle budget, as evidenced by the steepness of their performance decline curves. We interpret this behavior as an *inductive* Lipschitz regulation bias that corresponds to how suboptimal budget allocations (i.e., greater deviation from the oracle estimate) affect each architecture's ability to self-regulate local Lipschitzness. Observations for other architectures, including SIREN and Gauss, are provided in Appendix G.5.

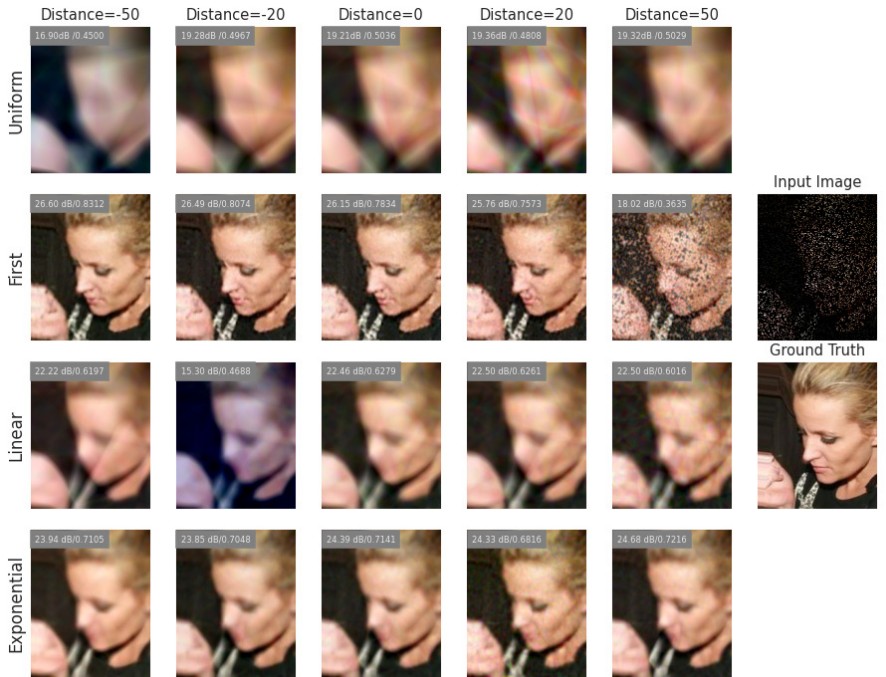

Figure 6: Qualitative examples from image inpainting experiments on the CelebA dataset using a Lipschitz FFN with $K_{min} = 0.5$ and SLL-normalization layers.

## 6  PRACTICAL GUIDELINES FOR LIPSCHITZ REGULARIZATION

While the experiments in section 5 demonstrate the efficacy of our framework in inverse problems, the core principles of the proposed Lipschitz framework are broadly applicable. In this section, we distill our findings into a generalized methodology for practitioners, addressing (1) deriving a principled global budget $K$, and (2) selecting an optimal allocation strategy.

**Strategies for Estimating the Budget $K$:**   We propose three distinct methods for estimating a Lipschitz budget, using available information:

- *Domain-driven estimate:* Beyond the tissue stretch constraints used in our experiments, domain-specific priors (e.g., max cardiac contraction) or intensity bounds can serve as interpretable upper bounds. For instance, in CT reconstruction, the maximal plausible gradient between adjacent voxels, such as the transition from air to tissue in Hounsfield Units, can directly be used to set an informed budget $K$ relative to spatial resolution.
- *Data-driven estimate:* When representative samples are available (e.g., high-resolution reference images for super-resolution tasks), they may serve as a meaningful reference signal for the proposed oracle to estimate Lipschitz budget as described in C. A dataset of representative images can quantify local variations, effectively transferring the smoothness prior from the observed domain to guide budget selection.
- *Signal-theoretic estimate:* Lacking strong priors or reference data, we recommend conservative estimation based on signal processing fundamentals (see Appendix C). Known bandlimits or sampling rates, such as those standard in audio (44.1 kHz) or electrocardiograms ($\approx$150 Hz), provide robust, noise-suppressing baselines for regularization.

**Strategies for Allocating the Budget $K$:** Distributing the global budget $K$ remains an open challenge (Gouk et al., 2021). We recommend treating allocation as a hyperparameter search centered on network expressivity. Specifically, practitioners should analyze the performance of different allocation strategies introduced in Section 3.2 with respect to $K_{min}$, i.e., the minimum imposed Lipschitz bound of the networks' components. This allows for a systematic exploration of the trade-off between rigid, uniform regularization and flexible, non-uniform budget distributions.

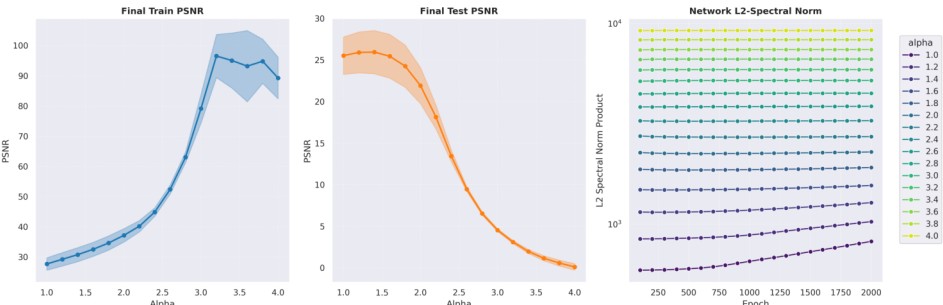

Figure 7: Visualization of the upper induced Lipschitz bound of the method proposed by Yeom et al. (2024) for SIREN. Scaling the initialization leads to direct scaling of the Lipschitz bounds of the linear layers, which allows the network to increase its capacity to overfit high-frequency content. We report layer-wise spectral norms for the experiment in subsection G.3.

# 7 HOW CAN LIPSCHITZ THEORY PROVIDE A NOVEL PERSPECTIVE?

So far, we have employed Lipschitz regularization as a measure to improve smoothness in learned neural representations, while setting a global constraint on allowed perturbations, thus directly guiding implicit regularization strength. In this section, we want to highlight the appeal of using Lipschitz theory to explain concurrent advances in neural field architectures, and provide an outlook for the community on how Lipschitz theory may complement and even go beyond NTK literature (Jacot et al., 2018; Yüce et al., 2022) and Fourier analysis (Benbarka et al., 2022; Ramasinghe et al., 2023) in providing a unifying paradigm. Recent work by Yeom et al. (2024) shows that scaling the initial and hidden layer weights of SIREN networks by a constant factor $\alpha$ improves both accuracy and convergence speed. While their explanation relies on optimization and NTK theory, we argue that Lipschitz theory offers a complementary perspective.

To test this, we performed an inpainting experiment with a three-layer, 256-neuron SIREN ($\omega = 30$), trained on 25% of the pixels and evaluated using PSNR on training and test sets. Following Yeom et al. (2024), we uniformly scaled the first and hidden layer weights by $\alpha \in [1.0, 4.0]$ in steps of 0.2. For a standard SIREN layer $f(x) = \sin(\omega(\alpha \boldsymbol{W} \boldsymbol{x} + \boldsymbol{b}))$, the induced Lipschitz constant is $\mathrm{Lip}(f) = |\omega \alpha \|\boldsymbol{W}\||$, showing that the upper Lipschitz bound scales linearly with $\alpha$. Empirical spectral analysis of the scaled layers and the full network confirms this relationship. Scaling increases the spectral norm, enabling the network to capture higher-frequency components and overfit more strongly. Moreover, layer-wise analysis shows that with small $\alpha$, the Lipschitz bound grows during training, while with large $\alpha$ it remains constant, indicating a self-regulating capacity when weights are sufficiently scaled.

DISCUSSION AND CONCLUSION:

In this study, we proposed a shift in how Lipschitz regularization is applied to INRs: from a rigid, global constraint to a more flexible Lipschitz budget. Using an extensive set of experiments from shape representation, lung registration, to image inpainting and super-resolution, we have demonstrated that successful Lipschitz regularization critically depends on (i) selecting a meaningful global budget, and (ii) utilizing appropriate allocation strategies.

To achieve this, we first introduced a systematic framework for deriving task-specific budgets, informed by data, domain, or signal-theoretic properties. Second, we showed that non-uniform budget allocation enables balancing regularization with expressiveness under the same global budget. After demonstrating the proposed framework across different modalities and tasks, we have provided a practical guide for adapting it to novel problems and applications.

In future work, we aim to extend task-specific budgets to novel domains and explore how budget allocation might generalize within and beyond spectral regularization.

ACKNOWLEDGEMENTS:

We thank Patricia Pauli for helpful discussions. This work is funded by the Munich Center for Machine Learning. Julian McGinnis and Mark Mühlau are supported by Bavarian State Ministry for Science and Art (Collaborative Bilateral Research Program Bavaria – Quebec: AI in medicine, grant F.4-V0134.K5.1/86/34). Suprosanna Shit is supported by the UZH Postdoc Grant (K-74851-03-01). Suprosanna Shit and Bjoern Menze acknowledge support by the Helmut Horten Foundation. This work was supported under project ID a135 as part of the Swiss AI Initiative, through a grant from the ETH Domain and computational resources provided by the Swiss National Supercomputing Centre (CSCS) under the Alps infrastructure.

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

## A  STATEMENTS

**Reproducibility Statement:**   The complete implementation, including data preprocessing, model training, and evaluation scripts, is available at `https://lipschitz-inrs.github.io/`. We further describe details regarding hyperparameters in Appendix D.

**Ethics Statement:**   Our research advances the theoretical foundations of implicit neural representations through novel regularization techniques. While our work focuses on fundamental methodological contributions without direct harmful applications, we recognize that implicit neural representations can potentially be applied to sensitive domains. We encourage practitioners to carefully evaluate privacy, security, and fairness implications when deploying such models in real-world applications, particularly those involving personal or sensitive data.

**Usage of Large Language Models:**   We utilized large language models as auxiliary tools during various phases of this paper while maintaining full responsibility for validating all outputs. Specifically, LLMs assisted with literature search, writing refinement, ideation and brainstorming during the initial development stage and brain-storming, and were utilized during the implementation of source code and and visualization utilities. All LLM-generated content was thoroughly reviewed and modified to ensure accuracy and appropriateness.

## B  ADDITIONAL EXPERIMENT: SINGLE-IMAGE SUPER-RESOLUTION

We study the applicability of the oracle estimate and the Lipschitz budget allocation for the task of single-image super-resolution, following the approach and experimental setup presented in Saragadam et al. (2023).

**Definition:**  Let us consider single-image super-resolution (SISR) as the task of reconstructing a super-resolved, high-resolution image $\hat{I}_{HR}$ from the observations of a low-resolution image $I_{LR}$. Here, the goal is to recover high-frequency details while reconstructing an image that is also consistent with $I_{LR}$.

Given an input coordinate $(x, y) \in \Omega$, where $\Omega$ is the continuous 2D image domain, we define $\Omega_{LR}$ and $\Omega_{HR}$ as the coordinate grids for $I_{LR}$ and $I_{HR}$ respectively. Following Saragadam et al. (2023), we train the INR on the full, high-resolution grid $\Omega_{HR}$, and compute the pixel-wise loss between the degraded, downsampled intensities $\mathcal{D}(f_\theta(\Omega_{HR}))$ and $I_{LR}$, where in our experiments, $\mathcal{D}$ represents a downsampling operator with a scale factor of four.

**Budget estimate:**  In light of super-resolution applications, we would like to discuss two useful oracles.

- Conservative estimate: Assuming that a representative set of natural images is not available for deriving a signal-based estimate, we can set the conservative, upper Lipschitz bound by considering the worst-case Lipschitz constant. (c.f. section C

- Signal-based estimate: For the sake of evaluation, let us consider that we have access to both the LR and HR images, and that we can measure the estimate directly using our proposed signal-based estimate (c.f. section C).

**Experiment:**  For the SISR experiments, we use the Set14 dataset (Zeyde et al., 2010) and train several common INR architectures, including SIREN (Sitzmann et al., 2020), FFNs (Tancik et al., 2020), and Gaussian-activated INRs (Ramasinghe & Lucey, 2022), using the allocation strategies presented earlier. We evaluate performance using standard metrics: PSNR ($\uparrow$), SSIM ($\uparrow$) (Wang et al., 2004), and LPIPS ($\downarrow$) (Zhang et al., 2018). The quantitative results are shown in Fig. 37, 38, 41 for spectral norm and $K_{min} = 0.25, 0.5, 1.0$, respectively. The statistical results are shown in Fig. 36, 39, 40 respectively. We also provide qualitative examples in Fig. 44.

# C   DATA DRIVEN ORACLE ESTIMATION FOR 2D RGB IMAGES

## C.1   SIGNAL-BASED ORACLE DERIVATION

Let $f : \Omega \subset \mathbb{R}^2 \to \mathbb{R}^3$ denote the RGB image-valued function with $f(\boldsymbol{x}, \boldsymbol{y}) = \big(f_R(\boldsymbol{x}, \boldsymbol{y}), f_G(\boldsymbol{x}, \boldsymbol{y}), f_B(\boldsymbol{x}, \boldsymbol{y})\big)^\top$. With Euclidean norms on input and output, the (global) Lipschitz constant of $f$ is

$$L = \inf\big\{\ell \geq 0 : \|f(\mathbf{p}_1) - f(\mathbf{p}_2)\|_2 \leq \ell \|\mathbf{p}_1 - \mathbf{p}_2\|_2 \ \forall \mathbf{p}_1, \mathbf{p}_2 \in \Omega\big\}.$$

If $f$ is $C^1$, then

$$L = \sup_{(\boldsymbol{x}, \boldsymbol{y}) \in \Omega} \big\|J_f(\boldsymbol{x}, \boldsymbol{y})\big\|_2, \qquad J_f(\boldsymbol{x}, \boldsymbol{y}) \in \mathbb{R}^{3 \times 2},$$

where $J_f$ is the Jacobian

$$J_f(\boldsymbol{x}, \boldsymbol{y}) = \begin{bmatrix} \partial_x f_R & \partial_y f_R \\ \partial_x f_G & \partial_y f_G \\ \partial_x f_B & \partial_y f_B \end{bmatrix}, \qquad \|J\|_2 = \sigma_{\max}(J).$$

Equivalently, if we write

$$J_f(\boldsymbol{x}, \boldsymbol{y})^\top J_f(\boldsymbol{x}, \boldsymbol{y}) = \begin{bmatrix} a & b \\ b & d \end{bmatrix}, \quad \text{with} \ a = \sum_{c \in \{R,G,B\}} (\partial_x f_c)^2, \ d = \sum_c (\partial_y f_c)^2, \ b = \sum_c (\partial_x f_c)(\partial_y f_c),$$

then

$$\|J_f(\boldsymbol{x}, \boldsymbol{y})\|_2 = \sqrt{\lambda_{\max}(J_f^\top J_f)} = \sqrt{\tfrac{1}{2}\Big(a + d + \sqrt{(a-d)^2 + 4b^2}\Big)}.$$

**Discrete estimation on a rectangular grid.**   Let the image be sampled on a grid $\{(x_j, y_i)\}$ with spacings $\Delta x$ and $\Delta y$. Using centered finite differences for interior points (with one–sided differences at the boundary),

$$\partial_x f_c(x_j, y_i) \approx \frac{f_c(x_{j+1}, y_i) - f_c(x_{j-1}, y_i)}{2\,\Delta x}, \qquad \partial_y f_c(x_j, y_i) \approx \frac{f_c(x_j, y_{i+1}) - f_c(x_j, y_{i-1})}{2\,\Delta y},$$

where $\Delta x \approx \frac{2}{(W-1)}$ and $\Delta y \approx \frac{2}{(H-1)}$ for an $H \times W$ size image assuming coordinate range in $[-1, 1]$. Define $a_{ij}, b_{ij}, d_{ij}$ from these discrete partials as above and set

$$\sigma_{ij} = \sqrt{\tfrac{1}{2}\Big(a_{ij} + d_{ij} + \sqrt{(a_{ij} - d_{ij})^2 + 4b_{ij}^2}\Big)}.$$

Our spectral (Euclidean) Lipschitz estimator is then

$$\widehat{L}_{\text{est}} = \max_{i,j} \sigma_{ij}.$$

## C.2   NOISE SENSITIVITY ANALYSIS FOR SIGNAL-BASED ORACLE

Following the common denoising setup in Saragadam et al. (2023), we evaluate the robustness of the signal-based oracle estimator under simulated photon shot noise. We vary the peak photon counts $\gamma \in \{30, 90, 120\}$ with a fixed readout noise of $\sigma = 2$, utilizing examples of our inpainting dataset (CelebA).

As shown in the plots, the oracle estimator is only moderately sensitive to noise. Although noise induces a slight shift in the gradient distribution, particularly in the tail, leading to an increased maximum estimated Lipschitz constant, the overall distributional structure remains largely preserved. This effect becomes more pronounced at high noise levels (e.g., $\gamma = 30$). In such cases, where noise

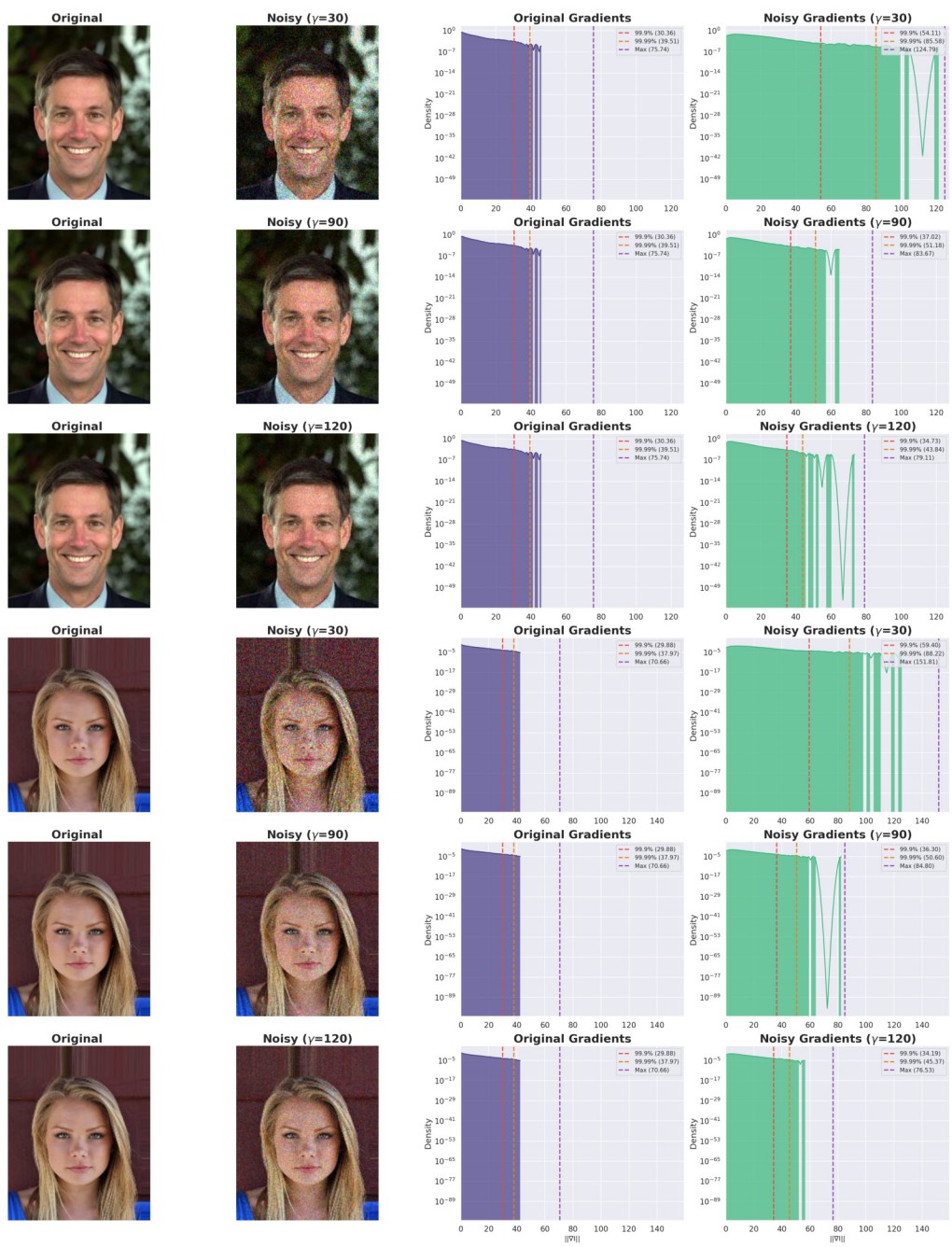

Figure 8: Sensitivity Analysis to simulated photon shot noise of various levels. We recommend switching to percentile-based estimates in noisy images instead of utilizing the absolute maximum estimated Lipschitz constant.

artifacts distort the extreme values, percentile-based thresholds provide a robust alternative to relying on the absolute maximum. In particular, we find that the 99.9% and 99.99% percentiles reduce the gap between estimates obtained from the original and noisy images.

### C.3 CONSERVATIVE ORACLE ESTIMATE:

In applications where access to representative samples is not possible, or the sample-based estimates exhibit high variability, we recommend considering a conservative global upper bound for the Lipschitz constant that can be set from the input sampling resolution and the output amplitude range.

Let $f_s$ be the sampling frequency, $f_N = f_s/2$ the Nyquist frequency, $R$ the peak-to-peak output range, and $C$ the number of output channels. For a continuous, bandlimited signal (worst-case sinusoid at $f_N$) with amplitude $A = R/2$, the derivative magnitude per channel is bounded by

$$\left| \frac{d}{dt} f_c(t) \right| \le 2\pi f_N A = \frac{\pi}{2} R f_s.$$

Bounding the operator norm of the Jacobian by its Frobenius norm yields the conservative global Lipschitz upper bound:

$$L \le \sqrt{C} \frac{\pi}{2} R f_s = \sqrt{C} \pi R f_N.$$

Let $R$ be the output range and $s_x, s_y$ the grid spacings along $x$ and $y$. For each channel $c$,

$$|\partial_x f_c| \le \frac{\pi}{2} \frac{R}{s_x}, \qquad |\partial_y f_c| \le \frac{\pi}{2} \frac{R}{s_y}.$$

Let us define:

$$a = \sum_c (\partial_x f_c)^2 \le C \frac{\pi^2}{4} \frac{R^2}{s_x^2}, \qquad d = \sum_c (\partial_y f_c)^2 \le C \frac{\pi^2}{4} \frac{R^2}{s_y^2}.$$

Using the norm inequality $\|J_f\|_2 \le \|J_f\|_F = \sqrt{a+d}$, the global bound becomes:

$$L \le \frac{\pi}{2} R \sqrt{C \left( \frac{1}{s_x^2} + \frac{1}{s_y^2} \right)}.$$

In the isotropic case, where x-/y- spacing are identical, i.e. $s_x = s_y = s$, this simplifies to

$$L \le \frac{\pi}{2} R \frac{\sqrt{2C}}{s}.$$

## D DETAILS REGARDING EXPERIMENTS

### D.1 SHAPES

For all shape experiments, we used the data from the Common 3D Test Model s[2], namely *armadillo, beast, bimba, cow, homer, ogre, spot, stanford-bunny, suzanne, teapot*. Following (Davies et al., 2020), we employ an 8-layer multi-layer perceptron for all experiments with 256 hidden dimensions, and no skip connections. We train all models for 1000 epochs, with a learning rate of $5e - 4$, a batch size of 16384, and a combination of SDF loss $\lambda = 1$, normal loss $\lambda = 0.05$, and eikonal loss $\lambda = 0.05$ (Gropp et al., 2020). For 1-Lipschitz neural networks, we use the native spectral regularization implemented in PyTorch (Paszke et al., 2019), Björck normalized layers implemented in Serrurier et al. (2020) and SLL layers from Araujo et al. (2023). All 1-Lipschitz activations are implemented using Serrurier et al. (2020).

---

[2] https://github.com/alecjacobson/common-3d-test-models

## D.2 DEFORMABLE REGISTRATION

We use the repository of Wolterink et al. (2022) [3], and use the identical setup in terms of dataset and hyperparameters. To have a comparable setup, we restrict the registration, similarly to Wolterink et al. (2022) to the lung area. Since Wolterink et al. (2022) do not report the folding ratio, we complement the existing metrics computation by implementing the folding ratio within the masked region of Deepali (Schuh et al., 2025) and erode the deformation field by three voxels based on boundary effects imposed by the masked registration.

## D.3 INPAINTING

For the inpainting experiments, we choose 25 random images of the CelebA (resolution: 218x178) dataset (Liu et al., 2015), and use 4-layer MLPs with a hidden dimension of 256. We interpret positional encoding and Fourier Features as a non-learnable, fixed first layer, and thus allocate one layer less for all models employing an embedding, in comparison to e.g. SIREN. We keep the learning rate $l_r = 1e - 3$ and number of epochs $n_{epochs} = 5000$ constant across all experiments.

## D.4 SINGLE IMAGE SUPER-RESOLUTION:

For the super-resolution experiments, we select all images from the Set14 super-resolution dataset (Zeyde et al., 2010) and use 4-layer MLPs with a hidden dimension of 256. We interpret positional encoding and Fourier Features as a non-learnable, fixed first layer, and thus allocate one layer less for all models employing an embedding, compared to, e.g., SIREN. We keep the learning rate $l_r = 1e - 3$ and number of epochs $n_{epochs} = 5000$ constant across all experiments.

## D.5 WEIGHT SCALING FROM A LIPSCHITZ LENS

For this experiment, we choose 10 random images of the CelebA (resolution: 218x178) dataset (Liu et al., 2015), and use a 3-layer SIREN with a hidden dimension of 256. We scale $alpha$ in steps of 0.2 within the interval $[1, 4]$.

---

[3] https://github.com/MIAGroupUT/IDIR

# E ADDITIONAL DETAILS REGARDING LIPSCHITZ DERIVATION

## E.1 DERIVATION OF LIPSCHITZ IN INDIVIDUAL COMPONENTS

In this section, we detail our derivation of Lipschitz constants with respect to the results presented in subsection 3.1.

## E.2 POSITIONAL ENCODING

To compute the Lipschitz constant of the positional encoding function $\gamma(p)$, we must find the supremum of the norm of its derivative. Since the derivative's magnitude is constant, the supremum is simply this constant value.

The given positional encoding function $\gamma : \mathbb{R} \to \mathbb{R}^{2L}$ is defined as:

$$\gamma(p) = \big(\sin(2^0\pi p), \cos(2^0\pi p), \ldots, \sin(2^{L-1}\pi p), \cos(2^{L-1}\pi p)\big)$$

This function maps a scalar value $p$ to a vector of dimension $2L$.

The Lipschitz constant of a differentiable function is the supremum of the norm of its derivative. We first compute the derivative of each component of the vector $\gamma(p)$ with respect to $p$. For each integer $k \in \{0, 1, \ldots, L-1\}$, the derivatives are:

$$\frac{d}{dp}\sin(2^k\pi p) = 2^k\pi\cos(2^k\pi p)$$

$$\frac{d}{dp}\cos(2^k\pi p) = -2^k\pi\sin(2^k\pi p)$$

The derivative of the entire vector function, $\gamma'(p)$, is a vector composed of these derivatives:

$$\gamma'(p) = \big(2^0\pi\cos(2^0\pi p), -2^0\pi\sin(2^0\pi p), \ldots, 2^{L-1}\pi\cos(2^{L-1}\pi p), -2^{L-1}\pi\sin(2^{L-1}\pi p)\big)$$

The Lipschitz constant $K$ is given by $K = \sup_p \|\gamma'(p)\|_2$. We compute the squared Euclidean norm of the derivative vector $\gamma'(p)$:

$$\|\gamma'(p)\|^2 = \sum_{k=0}^{L-1}\left(\left(\frac{d}{dp}\sin(2^k\pi p)\right)^2 + \left(\frac{d}{dp}\cos(2^k\pi p)\right)^2\right)$$

$$= \sum_{k=0}^{L-1}\big((2^k\pi\cos(2^k\pi p))^2 + (-2^k\pi\sin(2^k\pi p))^2\big)$$

$$= \sum_{k=0}^{L-1}\big((2^k\pi)^2\cos^2(2^k\pi p) + (2^k\pi)^2\sin^2(2^k\pi p)\big)$$

Factoring out $(2^k\pi)^2$ and using $\cos^2(\theta) + \sin^2(\theta) = 1$:

$$\|\gamma'(p)\|^2 = \sum_{k=0}^{L-1}(2^k\pi)^2\big(\cos^2(2^k\pi p) + \sin^2(2^k\pi p)\big)$$

$$= \sum_{k=0}^{L-1}(2^k\pi)^2(1) = \pi^2\sum_{k=0}^{L-1}(2^k)^2 = \pi^2\sum_{k=0}^{L-1}4^k$$

The sum $\sum_{k=0}^{L-1}4^k$ is a finite geometric series with first term $a = 1$, common ratio $r = 4$, and $L$ terms. The sum is given by the formula $S_L = a\frac{r^L-1}{r-1}$:

$$\sum_{k=0}^{L-1}4^k = 1\cdot\frac{4^L-1}{4-1} = \frac{4^L-1}{3}$$

Substituting this result back into the expression for the squared norm:

$$\|\gamma'(p)\|^2 = \pi^2 \frac{4^L - 1}{3}$$

The Lipschitz constant $\text{Lip}(\gamma) = K$ is the square root of this value. Since the norm is constant, it is equal to its supremum:

$$K = \|\gamma'(p)\| = \sqrt{\pi^2 \frac{4^L - 1}{3}}$$

$$K = \pi \sqrt{\frac{4^L - 1}{3}}$$

Now, if we want to specifically set it, we can solve it for $L$ giving:

$$L = \log_4\left(3(K/\pi)^2 + 1\right) = \frac{\ln\left(3(K/\pi)^2 + 1\right)}{\ln 4}.$$

### E.3 RANDOM FOURIER FEATURES

Similarly, for a Random Fourier Feature (RFF) encoding, we consider

$$\gamma(v) = \left( \cos(2\pi b_1^\top v),\ \sin(2\pi b_1^\top v),\ \ldots,\ \cos(2\pi b_m^\top v),\ \sin(2\pi b_m^\top v) \right)^\top \in \mathbb{R}^{2m},$$

where $v \in \mathbb{R}^d$ and $b_j \in \mathbb{R}^d$.

The Lipschitz constant $K$ of $\gamma(v)$ is defined as the supremum of the spectral norm (largest singular value) of its Jacobian matrix $J_\gamma(v)$:

$$K = \sup_v \|J_\gamma(v)\|_2 = \sup_v \sigma_{\max}(J_\gamma(v)).$$

The Jacobian $J_\gamma(v)$ is a $2m \times d$ matrix. Its entries are given by the derivatives:

$$\frac{\partial}{\partial v} \cos(2\pi b_j^\top v) = -2\pi \sin(2\pi b_j^\top v)\, b_j^\top, \qquad \frac{\partial}{\partial v} \sin(2\pi b_j^\top v) = 2\pi \cos(2\pi b_j^\top v)\, b_j^\top.$$

The squared spectral norm is the largest eigenvalue of the Gram matrix $J_\gamma(v)^\top J_\gamma(v)$.

$$J_\gamma(v)^\top J_\gamma(v) = \sum_{j=1}^m \left( (-2\pi \sin(2\pi b_j^\top v)b_j)(-2\pi \sin(2\pi b_j^\top v)b_j^\top) + (2\pi \cos(2\pi b_j^\top v)b_j)(2\pi \cos(2\pi b_j^\top v)b_j^\top) \right)$$

$$= (2\pi)^2 \sum_{j=1}^m \left( \sin^2(2\pi b_j^\top v)b_j b_j^\top + \cos^2(2\pi b_j^\top v)b_j b_j^\top \right)$$

$$= (2\pi)^2 \sum_{j=1}^m \left( \sin^2(2\pi b_j^\top v) + \cos^2(2\pi b_j^\top v)\right)b_j b_j^\top$$

$$= (2\pi)^2 \sum_{j=1}^m b_j b_j^\top.$$

Since this result is independent of $v$, the Lipschitz constant is also independent of $v$.

$$K = \|J_\gamma(v)\|_2 = \sqrt{\lambda_{\max}(J_\gamma(v)^\top J_\gamma(v))} = 2\pi \sqrt{\lambda_{\max}\left( \sum_{j=1}^m b_j b_j^\top \right)},$$

where $\lambda_{\max}(A)$ denotes the largest eigenvalue of the matrix $A$.

More generally, if the encoding uses amplitudes $a_j$, i.e.

$$\gamma(v) = \left( a_1 \cos(2\pi b_1^\top v),\ a_1 \sin(2\pi b_1^\top v),\ \ldots,\ a_m \cos(2\pi b_m^\top v),\ a_m \sin(2\pi b_m^\top v) \right)^\top,$$

Then the Lipschitz constant generalizes to

$$K = 2\pi \sqrt{\lambda_{\max}\left(\sum_{j=1}^{m} a_j^2 b_j b_j^\top\right)}.$$

If the frequencies are sampled from a Gaussian distribution, $b_j \sim \mathcal{N}(0, \sigma^2 I_d)$, then $\sum_{j=1}^{m} a_j^2 b_j b_j^\top$ is a random matrix. The expected Lipschitz constant is then given by:

$$\mathbb{E}\left[K\right] = 2\pi \mathbb{E}\left[\sqrt{\lambda_{\max}\left(\sum_{j=1}^{m} a_j^2 b_j b_j^\top\right)}\right].$$

For common RFFs as proposed by Tancik et al. (2020), where all amplitudes are equal, $a_j = 1$, the expression becomes:

$$\mathbb{E}\left[K\right] = 2\pi \mathbb{E}\left[\sqrt{\lambda_{\max}\left(\sum_{j=1}^{m} b_j b_j^\top\right)}\right].$$

### E.4    LIPSCHITZ BUDGET ALLOCATION

We visualize different budget allocation strategies for the budget $K_B = 2$ as used in the deformable image registration experiment in 5.1

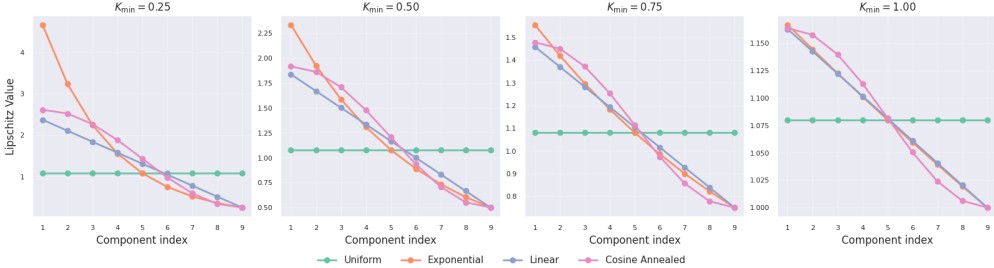

Figure 9: Exemplary Lipschitz budget allocation strategy for a budget $K_B = 2$ for a 9 component (5-layer) network.

## F    ADDITIONAL REGISTRATION BASELINES

To contextualize the deformable image registration results, we evaluate two additional baselines to benchmark the trade-off between target registration accuracy (TRE) and topological preservation (folding ratio).

While Lipschitz INRs are not specifically designed to avoid folding, but rather to model anatomically plausible transformations by constraining the maximum stretch, they remain particularly effective in these experiments and strike a good balance between TRE and the folding ratio. For the folding ratio, they outperform SIREN, with Jacobian regularization, in 8 out of 10 cases. Importantly, Lipschitz SIRENs remain robust across all cases and do not fluctuate as strongly as SIREN or NODEO. NODEO, as expected, yields the state-of-the-art folding ratio, but results in a higher TRE given that its transformations are strictly constrained, posing a challenge to the learning of the transformation.

Table 1: Comparison of TRE and Folding Ratio for SIREN, Lipschitz SIREN, and NODEO.

| | SIREN | | Lipschitz SIREN | | NODEO | |
|---|---|---|---|---|---|---|
| Case | TRE ($\downarrow$) | Folding Ratio ($\downarrow$) | TRE ($\downarrow$) | Folding Ratio ($\downarrow$) | TRE ($\downarrow$) | Folding Ratio ($\downarrow$) |
| 1 | $0.87 \pm 0.95$ | $2.43 \times 10^{-3}$ | $0.78 \pm 0.93$ | $2.13 \times 10^{-3}$ | $1.20 \pm 1.69$ | $6.61 \times 10^{-7}$ |
| 2 | $0.76 \pm 0.92$ | $3.46 \times 10^{-3}$ | $0.79 \pm 0.92$ | $3.54 \times 10^{-3}$ | $2.13 \pm 3.61$ | $2.94 \times 10^{-5}$ |
| 3 | $1.03 \pm 1.04$ | $4.63 \times 10^{-3}$ | $0.98 \pm 1.07$ | $4.84 \times 10^{-3}$ | $2.60 \pm 3.87$ | $1.08 \times 10^{-4}$ |
| 4 | $1.46 \pm 1.33$ | $4.96 \times 10^{-3}$ | $1.42 \pm 1.33$ | $4.50 \times 10^{-3}$ | $4.72 \pm 5.14$ | $4.82 \times 10^{-5}$ |
| 5 | $1.22 \pm 1.41$ | $4.01 \times 10^{-3}$ | $1.48 \pm 1.58$ | $3.64 \times 10^{-3}$ | $4.37 \pm 5.96$ | $2.77 \times 10^{-5}$ |
| 6 | $1.20 \pm 1.06$ | $1.93 \times 10^{-3}$ | $1.26 \pm 1.08$ | $1.92 \times 10^{-3}$ | $7.95 \pm 8.43$ | $7.54 \times 10^{-7}$ |
| 7 | $9.39 \pm 10.48$ | $5.07 \times 10^{-3}$ | $1.41 \pm 1.09$ | $1.80 \times 10^{-3}$ | $8.57 \pm 9.52$ | $2.22 \times 10^{-5}$ |
| 8 | $1.22 \pm 1.26$ | $2.65 \times 10^{-3}$ | $1.52 \pm 1.27$ | $2.45 \times 10^{-3}$ | $12.55 \pm 11.10$ | $5.63 \times 10^{-5}$ |
| 9 | $21.69 \pm 15.52$ | $7.21 \times 10^{-3}$ | $1.30 \pm 1.01$ | $9.90 \times 10^{-4}$ | $4.78 \pm 4.79$ | $2.65 \times 10^{-6}$ |
| 10 | $3.45 \pm 6.93$ | $3.68 \times 10^{-3}$ | $1.34 \pm 1.17$ | $1.63 \times 10^{-3}$ | $4.96 \pm 6.41$ | $5.00 \times 10^{-5}$ |

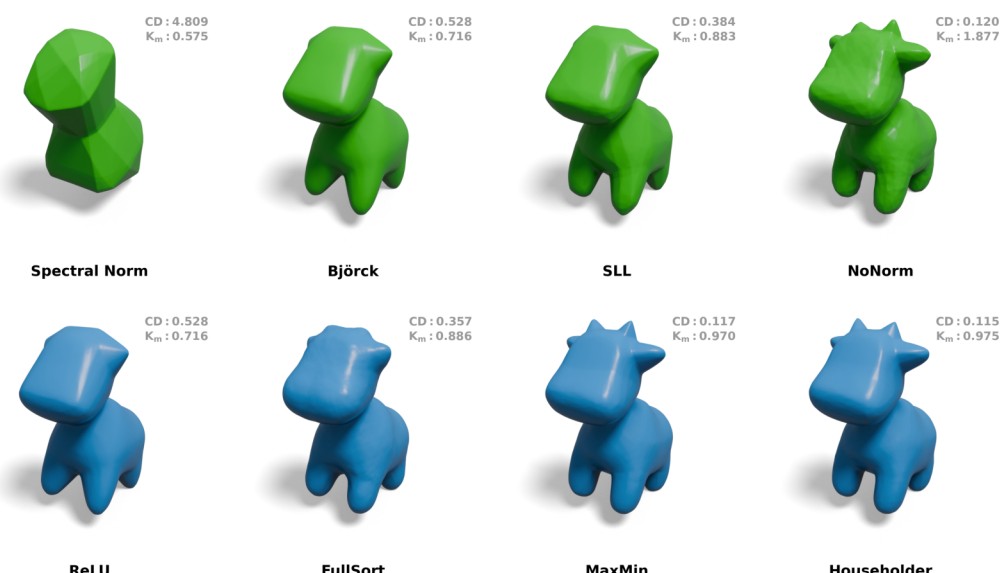

Figure 10: Learning *Spot* with different normalization techniques (first row) and different gradient-preserving activation functions in combination with a Björck normalized linear layer (second row) demonstrates that approaching the upper Lipschitz bound in SDFs correlates with perceptual quality. We report the CD, Chamfer Distance ($\downarrow$) and the empirically estimated Lipschitz constant $K_m$ for all reconstructions.

# G  ADDITIONAL RESULTS FOR EXPERIMENTS

## G.1  SHAPE EXPERIMENTS

## G.2  REGISTRATION EXPERIMENTS

We exemplarily visualize our registration setup in 14.

We ablate the importance of the lower Lipschitz bounds in the allocation experiments for deformable image registration in Fig. 15.

## G.3  SCALING WEIGHTS IN SIREN

We provide additional plots for the inpainting experiments in section 7.

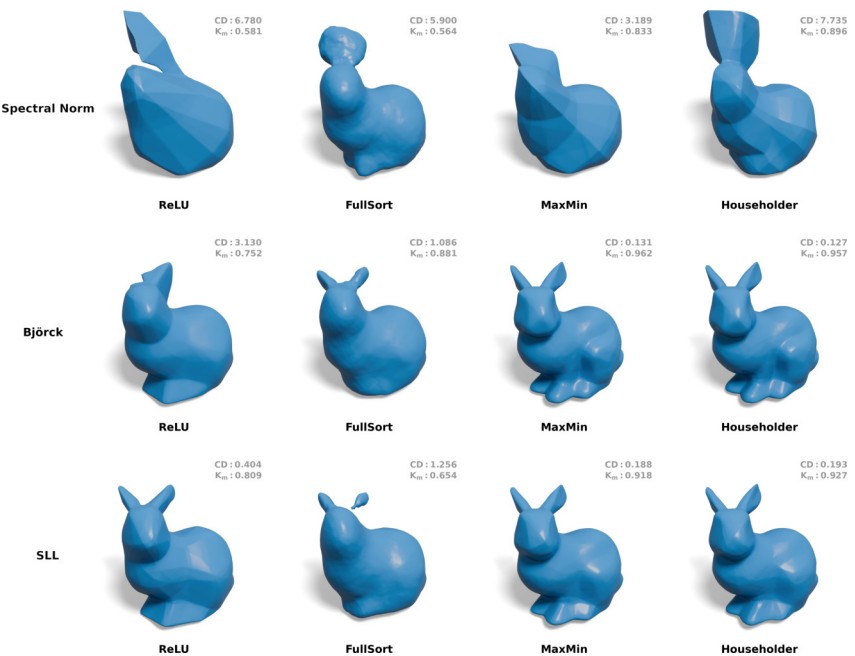

Figure 11: Learning *Stanford bunny* with different gradient-preserving activation functions (columns) in combination with different normalized linear layer (rows) demonstrates that approaching the upper Lipschitz bound in SDFs correlates with perceptual quality. We report the CD, Chamfer Distance ($\downarrow$) and the empirically estimated Lipschitz constant $K_m$ for all reconstructions.

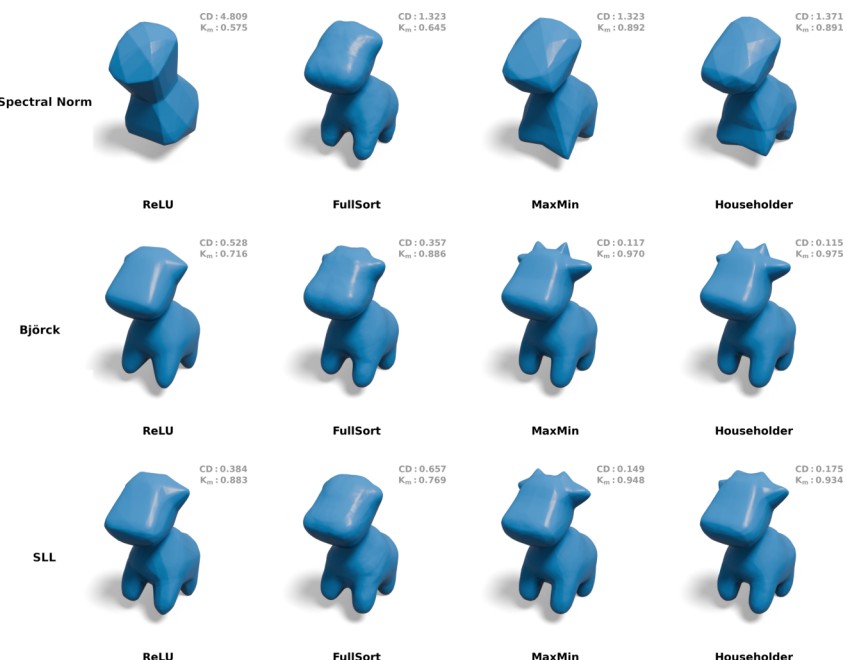

Figure 12: Learning *Spot* with different gradient-preserving activation functions (columns) in combination with different normalized linear layer (rows) demonstrates that approaching the upper Lipschitz bound in SDFs correlates with perceptual quality. We report the CD, Chamfer Distance ($\downarrow$) and the empirically estimated Lipschitz constant $K_m$ for all reconstructions.

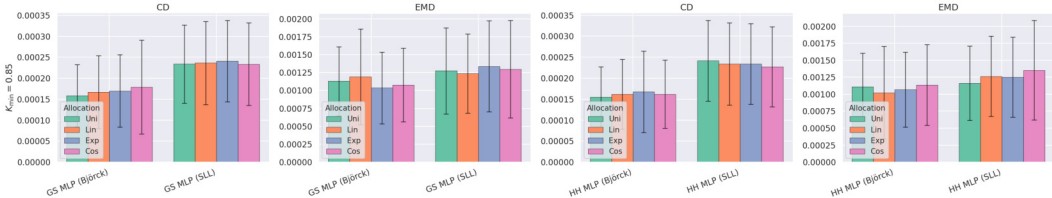

Figure 13: Visualization of the budget allocation experiment for SDFs using $k = 0.85$. Results similarly indicate no clear trend for improvements with non-uniform allocation strategies.

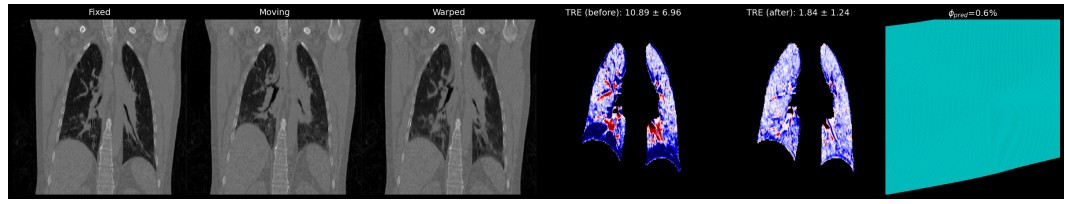

Figure 14: Visualization of warped deformation field and TRE before/after registration. Notably, Lipschitz INRs naturally enforce anatomically plausible transformations without other forms of auxiliary regularization, striking a good balance between registration accuracy and folding ratio.

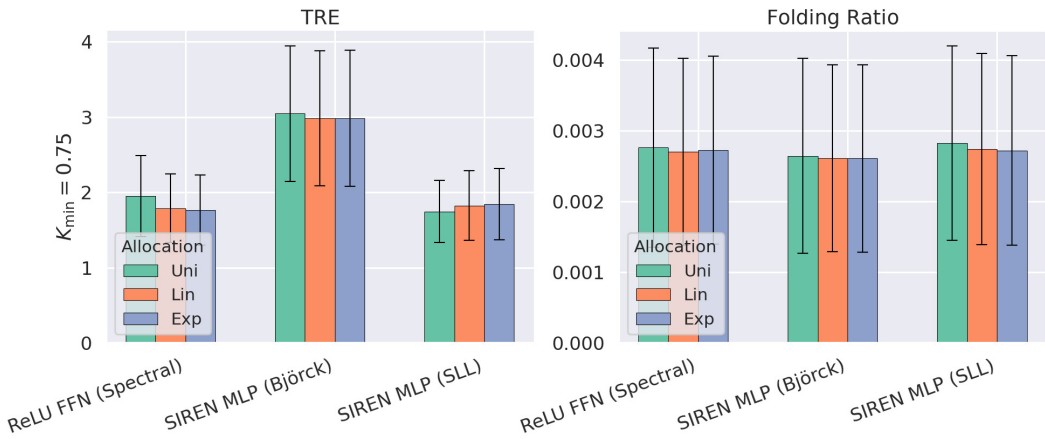

Figure 15: Visualization of the deformable image registration results of the budget allocation experiments using $K_{min} = 0.75$.

### G.4 STATISTICAL SIGNIFICANCE TEST

We perform a statistical significance test to determine the effect of different non-uniform allocation strategies against a uniform one. For the inpainting experiments, we select *distance-to-oracle* value at $-20, 0, 20$ and perform the Wilcoxon Signed rank test between three non-uniform (First, Linear, Exponential) with the uniform one. The results for Spectral Norm, Björck, and SLL normalization are presented in Fig. (31, 32, 35), (24, 27, 28),(19, 20, 23), respectively with different $K_{min}$ value. We employ the test for all three metrics, PSNR, SSIM, and LPIPS. We observe that non-uniform strategies consistently yield significant results ($p < 0.05$) compared to uniform strategies.

Similarly, we perform a Wilcoxon test in registration experiments to determine whether non-uniform allocation is statistically significant than uniform. The results are provided in Fig. 16. We observe

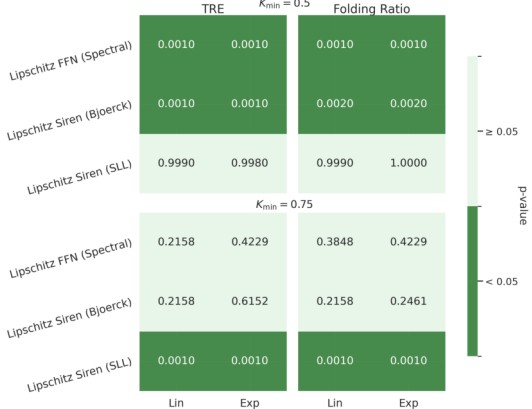

Figure 16: Heatmap for statistical significance test for registration experiment with $K_{min} = 0.5$ and $K_{min} = 0.75$. Green indicates that non-uniform allocation is statistically better ($p < 0.05$) than uniform allocation. We observe that non-uniform strategies offer significant results for both TRE and Folding Ratio in Lipschitz FFN (Spectral) and Lipschitz Siren (Björck) for $K_{min} = 0.5$ and in Lipschitz Siren (SLL) for $K_{min} = 0.75$.

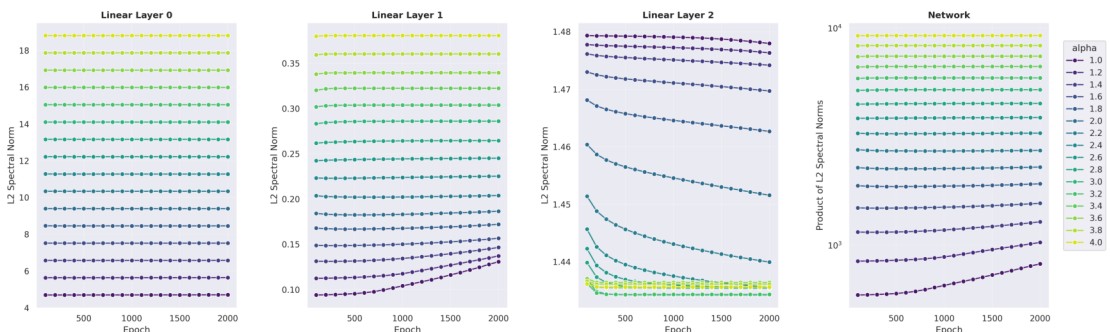

Figure 17: Visualization of the upper induced Lipschitz bound of the scaling initialization method proposed by Yeom et al. (2024) for SIREN. Scaling the initialization leads to direct scaling of the Lipschitz bounds of the scaled linear layers, which allows the network to increase its capacity to overfit high-frequency content. Please note that following (Yeom et al., 2024), we only scale the initial and hidden layers, but not the final layer (i.e. layer 2 in this case.)

that, for each architecture, a non-uniform strategy with an appropriate $K_{min}$ achieves significantly better results than uniform allocation.

We also provide similar statistically significant results on super-resolution experiments in Fig. 36,39,40 for different $K_{min}$ respectively. We observe that a non-uniform strategy yields statistically better results in most cases.

### G.5 ADDITIONAL INPAINTING RESULTS

Due to space constraints, we provide the remaining results of the budget allocation experiments for the inpainting application here. While FFNs demonstrate limited self-regulation capabilities, SIREN, for instance, exhibits superior self-regulation properties. We also provide statistical test results for each experiment setup. Further, we provide qualitative examples in Fig. 42 and 43.

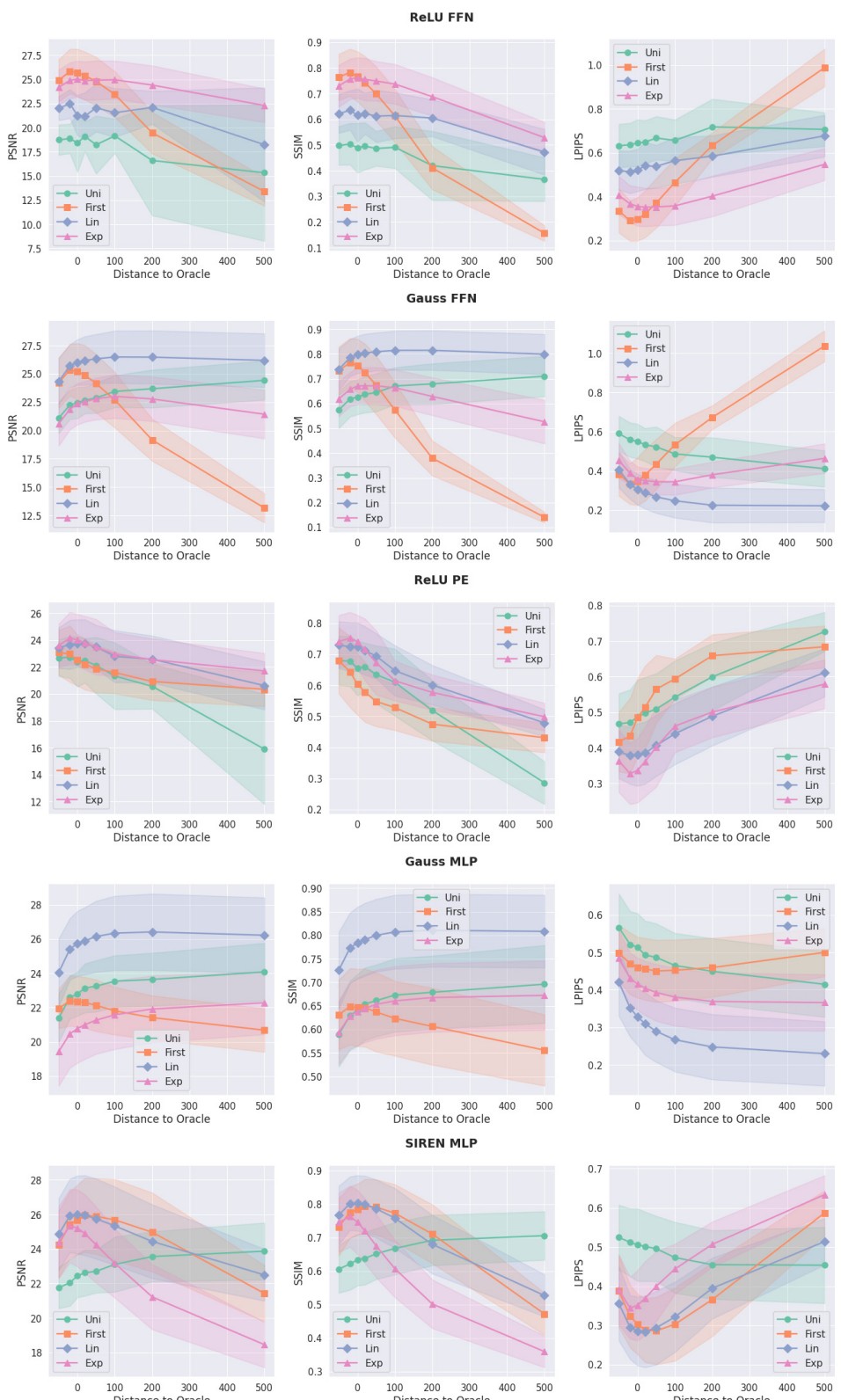

Figure 18: Budget Allocation for inpainting experiments for SLL with $K_{min} = 0.25$

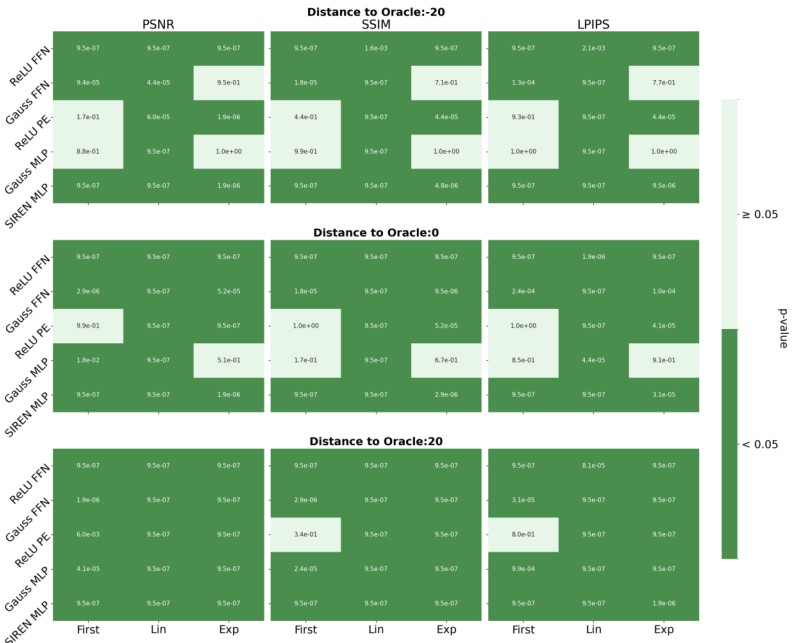

Figure 19: Heatmap for statistical significance test for inpainting experiment with SLL normalization with $K_{min} = 0.25$. Green indicates that non-uniform allocation is statistically better ($p < 0.05$) than uniform allocation. We observe that non-uniform strategies offer significant results in most cases across PSNR, SSIM, and LPIPS.

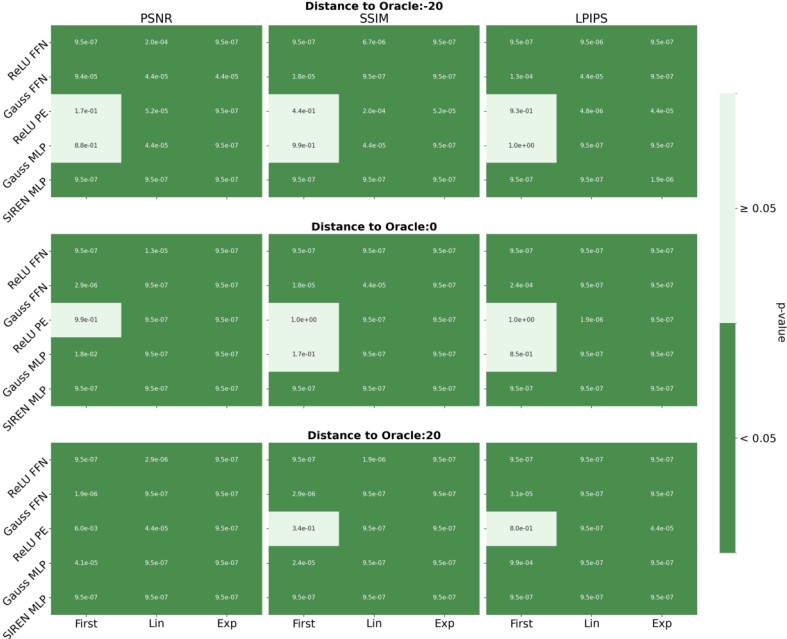

Figure 20: Heatmap for statistical significance test for inpainting experiment with SLL normalization with $K_{min} = 0.5$. Green indicates that non-uniform allocation is statistically better ($p < 0.05$) than uniform allocation. We observe that non-uniform strategies offer significant results in most cases across PSNR, SSIM, and LPIPS.

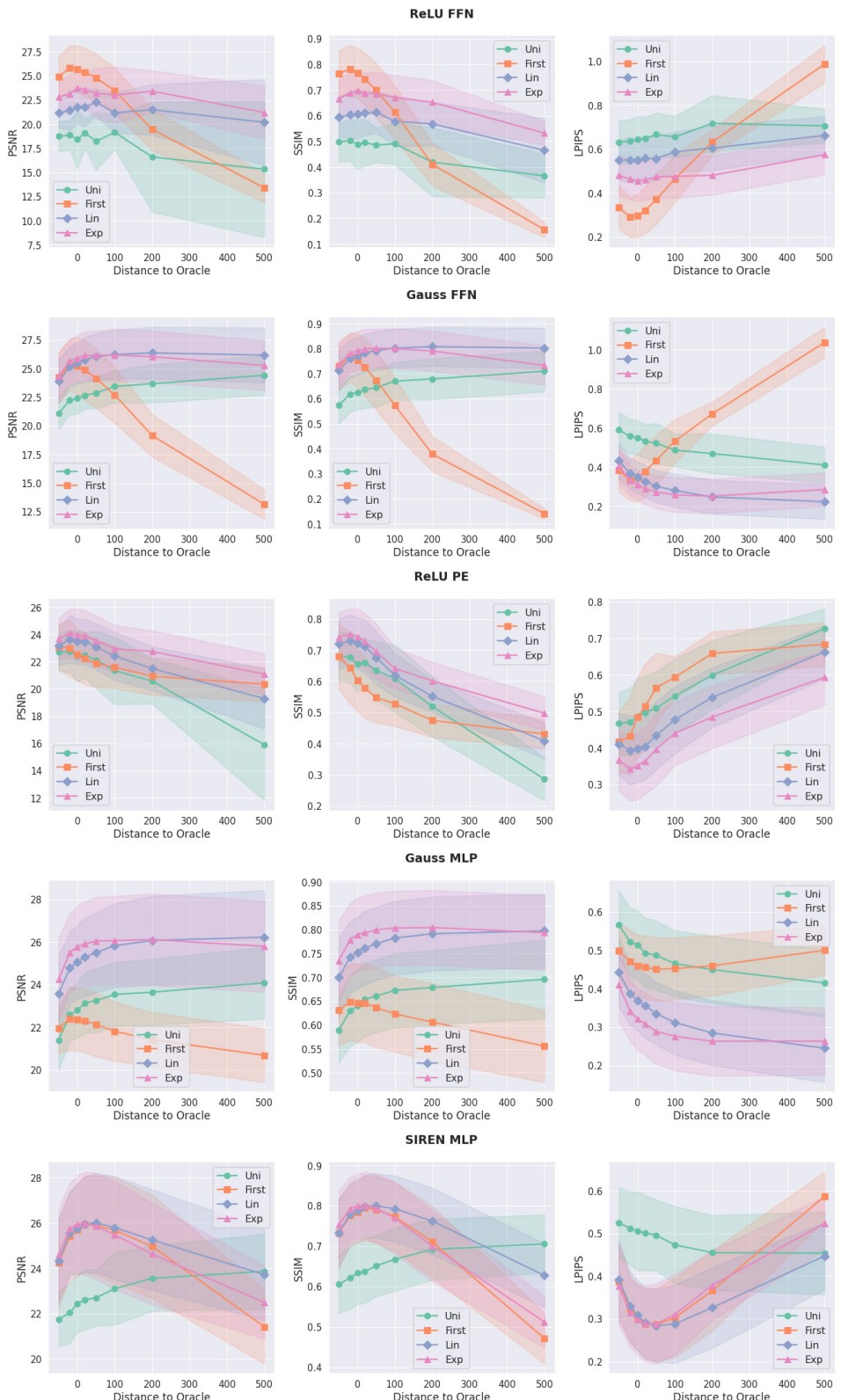

Figure 21: Budget Allocation for inpainting experiments for SLL with $K_{min} = 0.5$

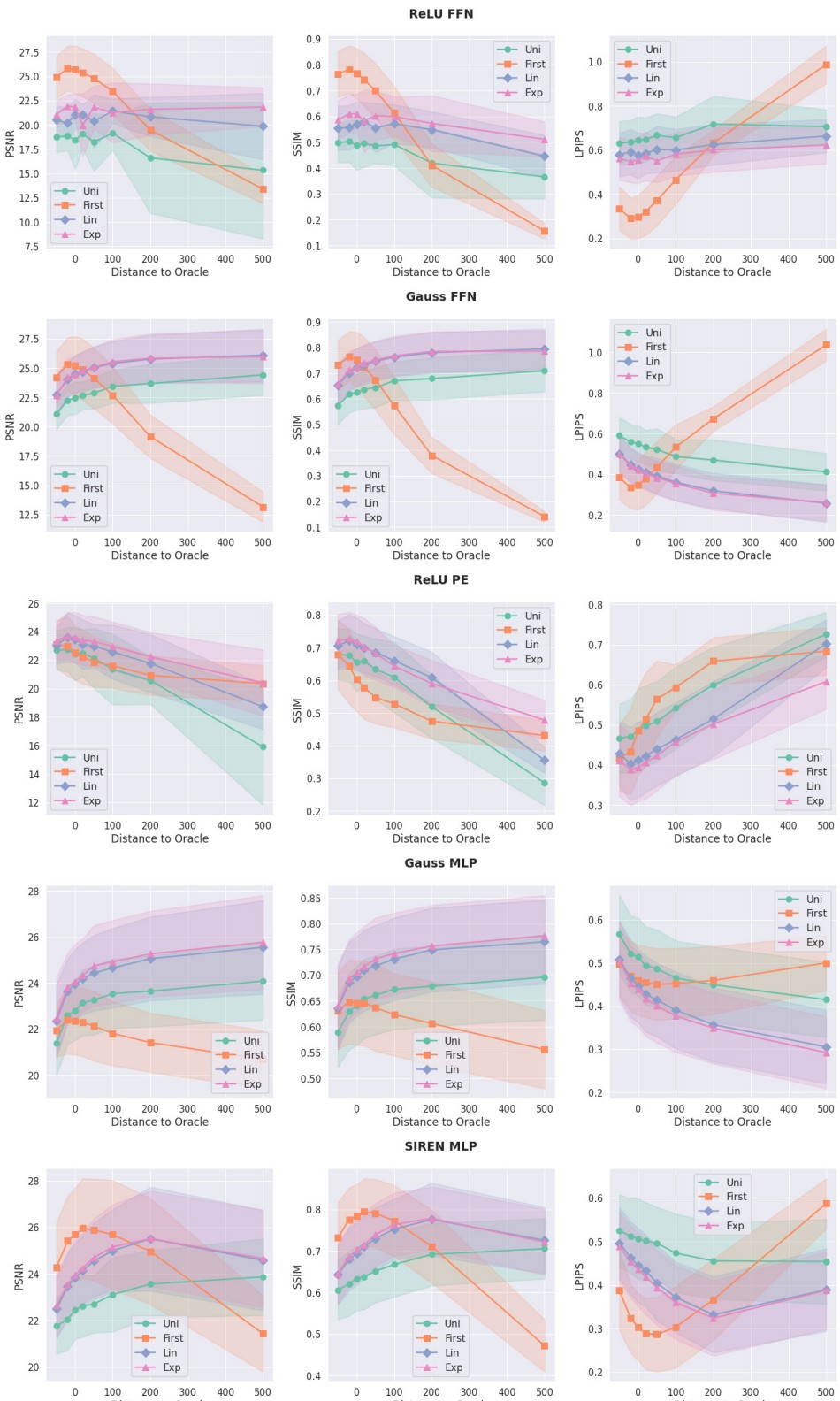

Figure 22: Budget Allocation for inpainting experiments for SLL with $K_{min} = 1.0$

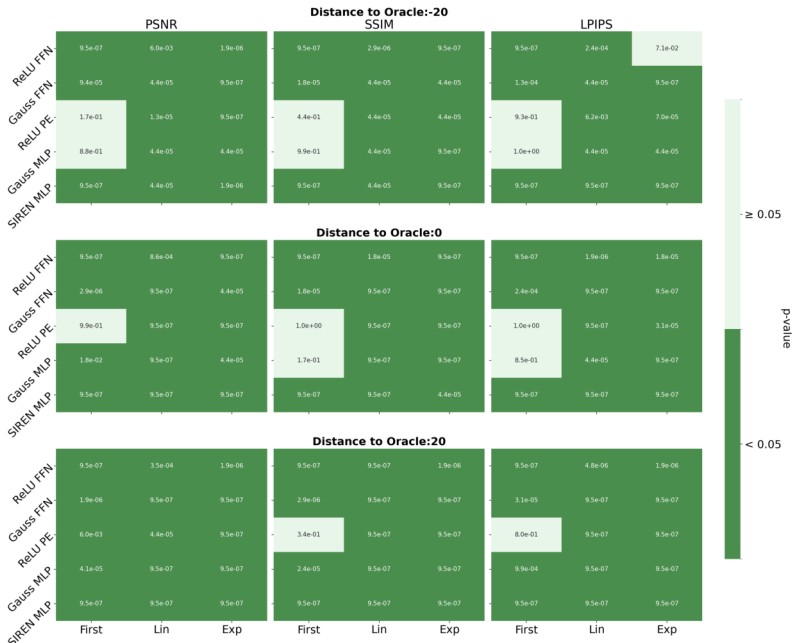

Figure 23: Heatmap for statistical significance test for inpainting experiment with SLL normalization with $K_{min} = 1.0$. Green indicates that non-uniform allocation is statistically better ($p < 0.05$) than uniform allocation. We observe that non-uniform strategies offer significant results in most cases across PSNR, SSIM, and LPIPS.

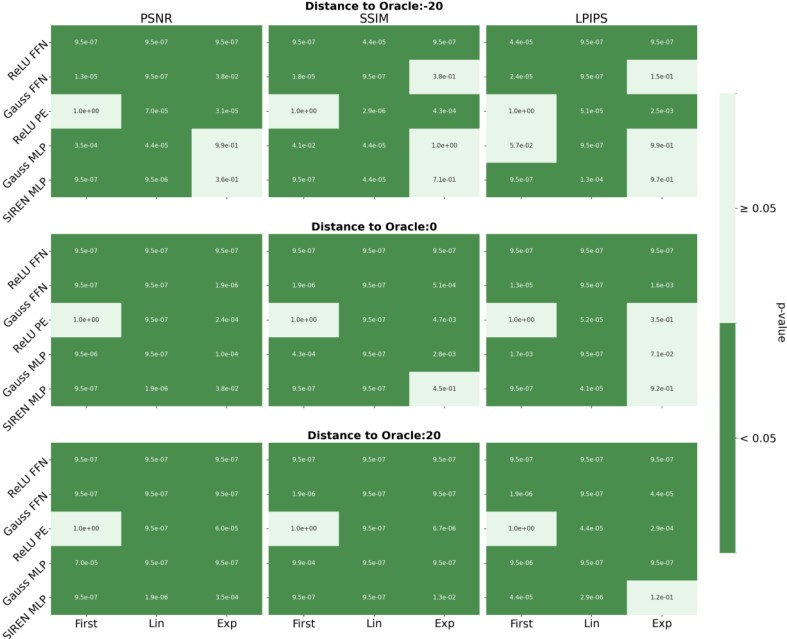

Figure 24: Heatmap for statistical significance test for inpainting experiment with Björck normalization with $K_{min} = 0.25$. Green indicates that non-uniform allocation is statistically better ($p < 0.05$) than uniform allocation. We observe that non-uniform strategies offer significant results in most cases across PSNR, SSIM, and LPIPS.

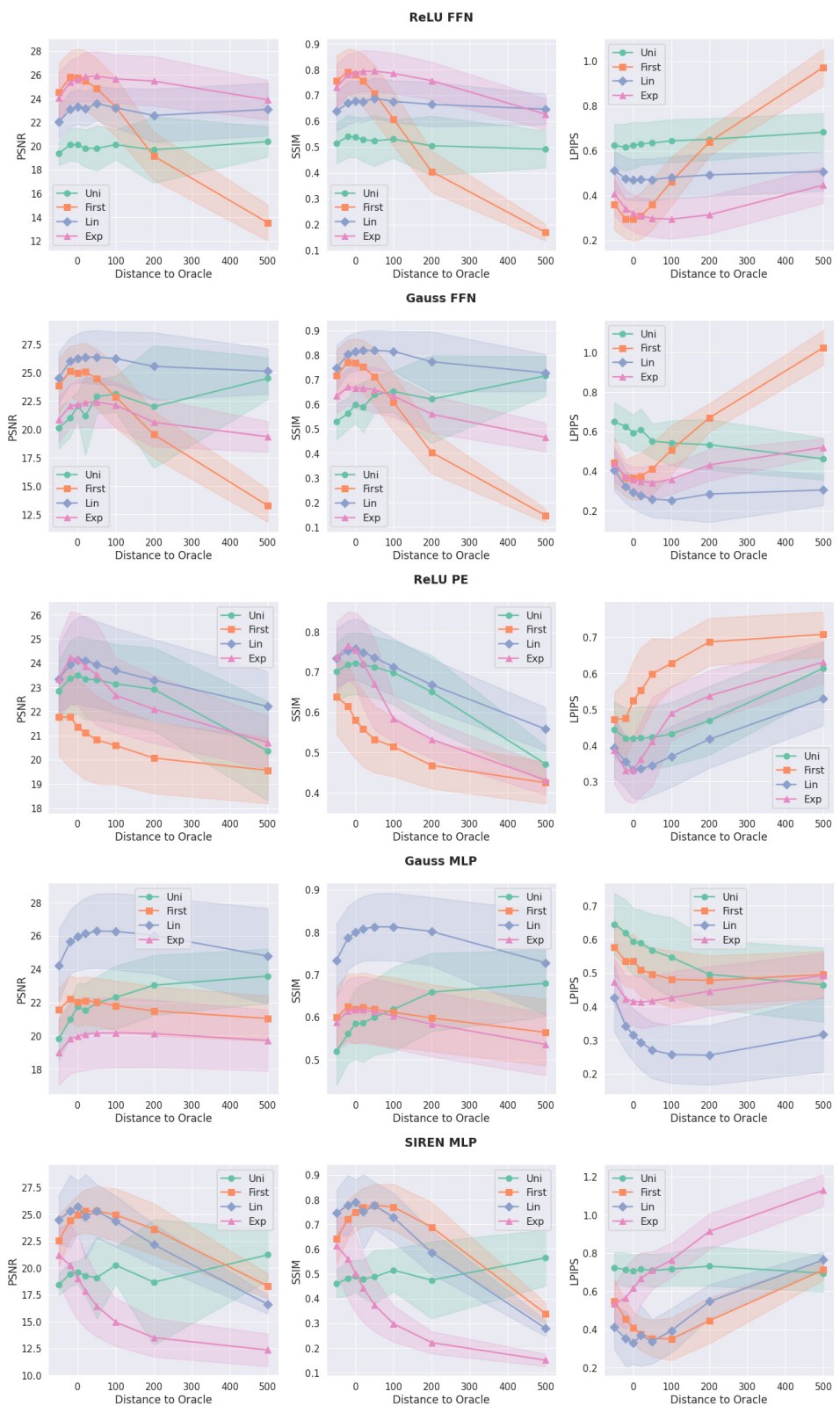

Figure 25: Budget Allocation for inpainting experiments for Björck with $K_{min} = 0.25$

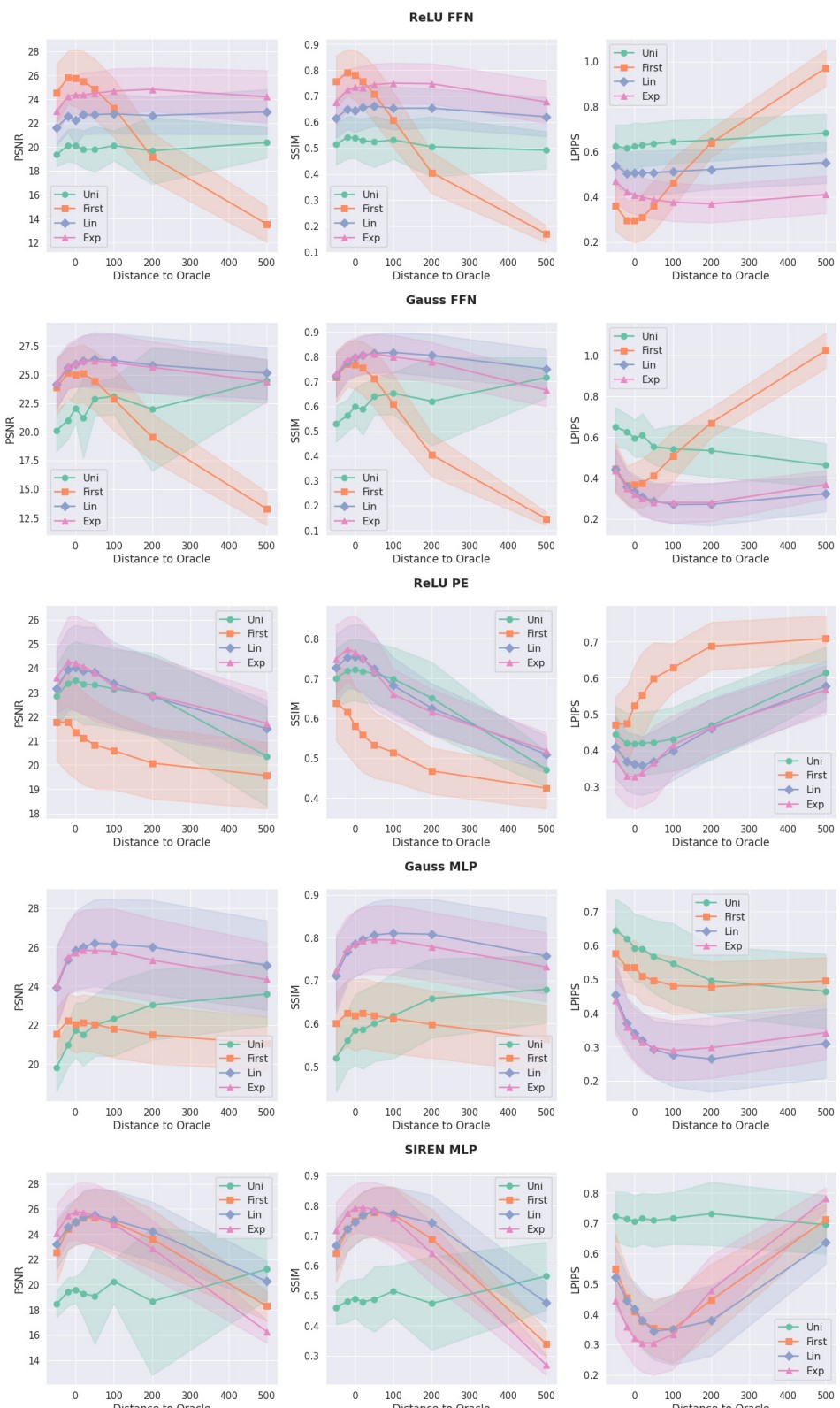

Figure 26: Budget Allocation for inpainting experiments for Björck with $K_{min} = 0.5$

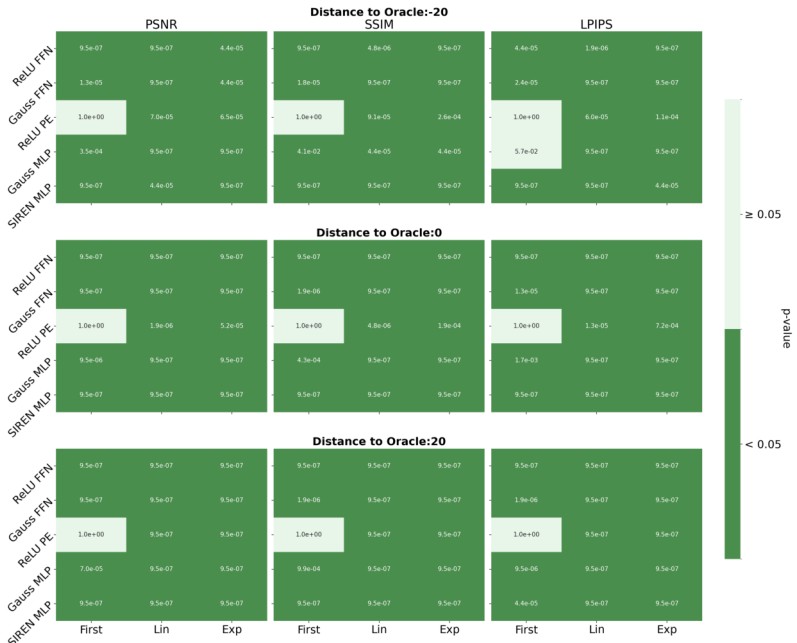

Figure 27: Heatmap for statistical significance test for inpainting experiment with Björck normalization with $K_{min} = 0.5$. Green indicates that non-uniform allocation is statistically better ($p < 0.05$) than uniform allocation. We observe that non-uniform strategies offer significant results in most cases across PSNR, SSIM, and LPIPS.

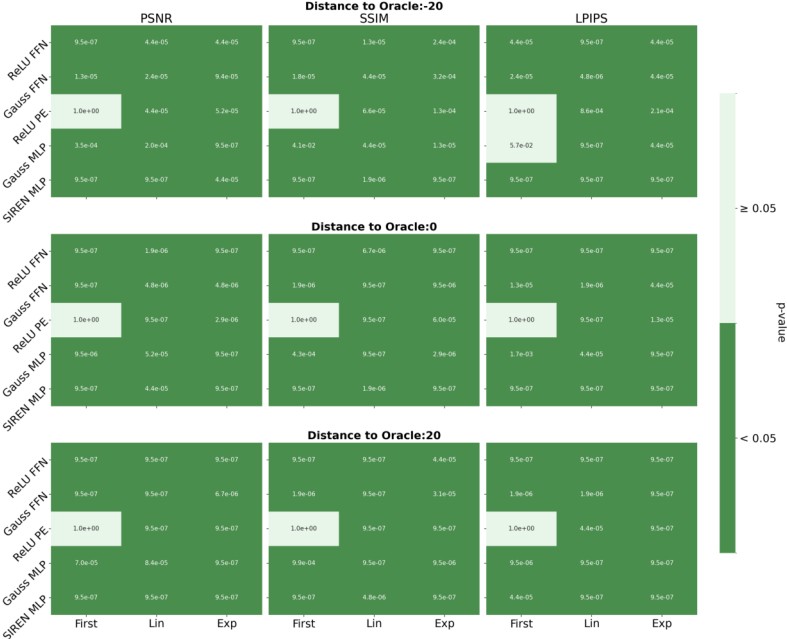

Figure 28: Heatmap for statistical significance test for inpainting experiment with Björck normalization with $K_{min} = 1.0$. Green indicates that non-uniform allocation is statistically better ($p < 0.05$) than uniform allocation. We observe that non-uniform strategies offer significant results in most cases across PSNR, SSIM, and LPIPS.

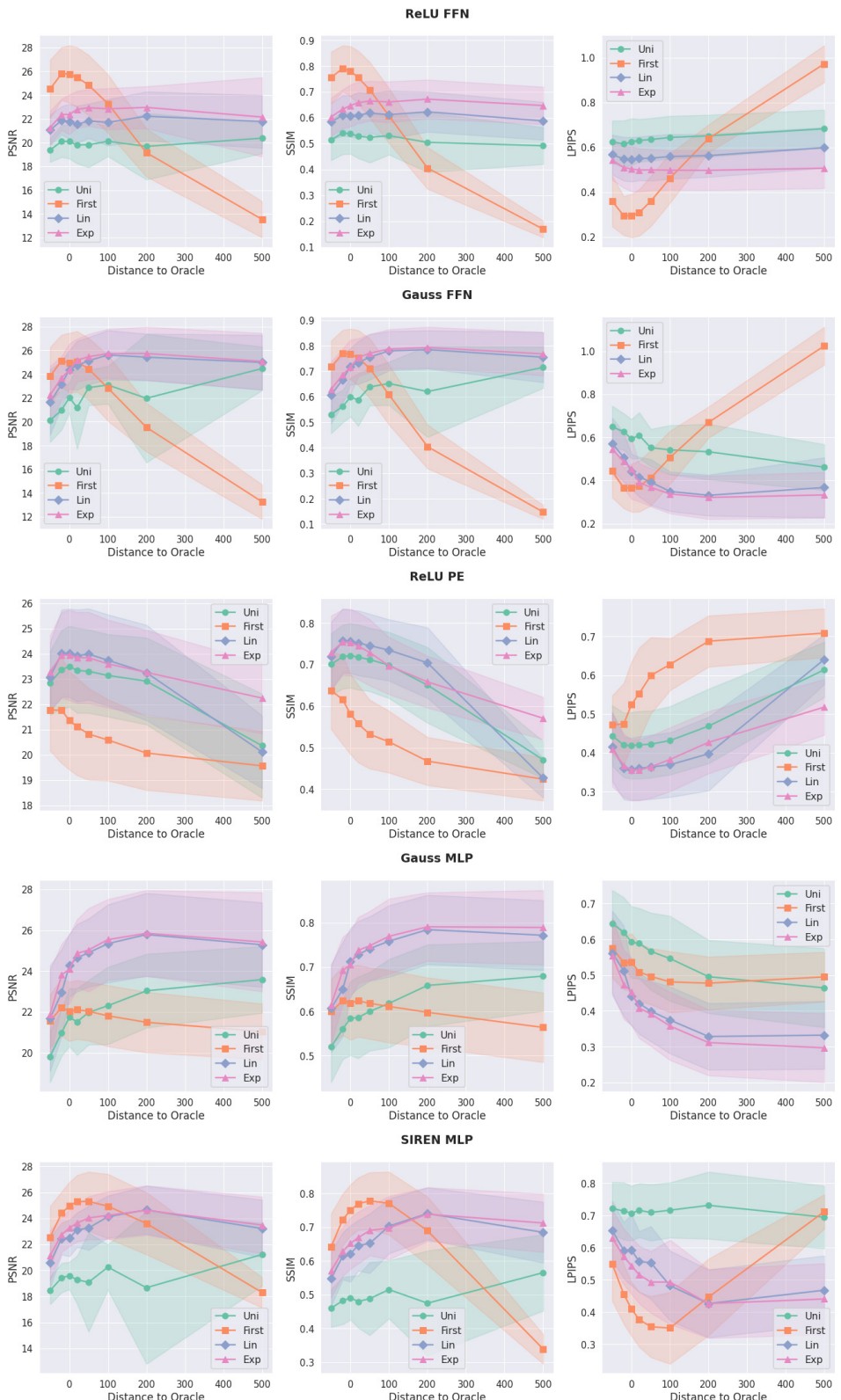

Figure 29: Budget Allocation for inpainting experiments for Björck with $K_{min} = 1.0$

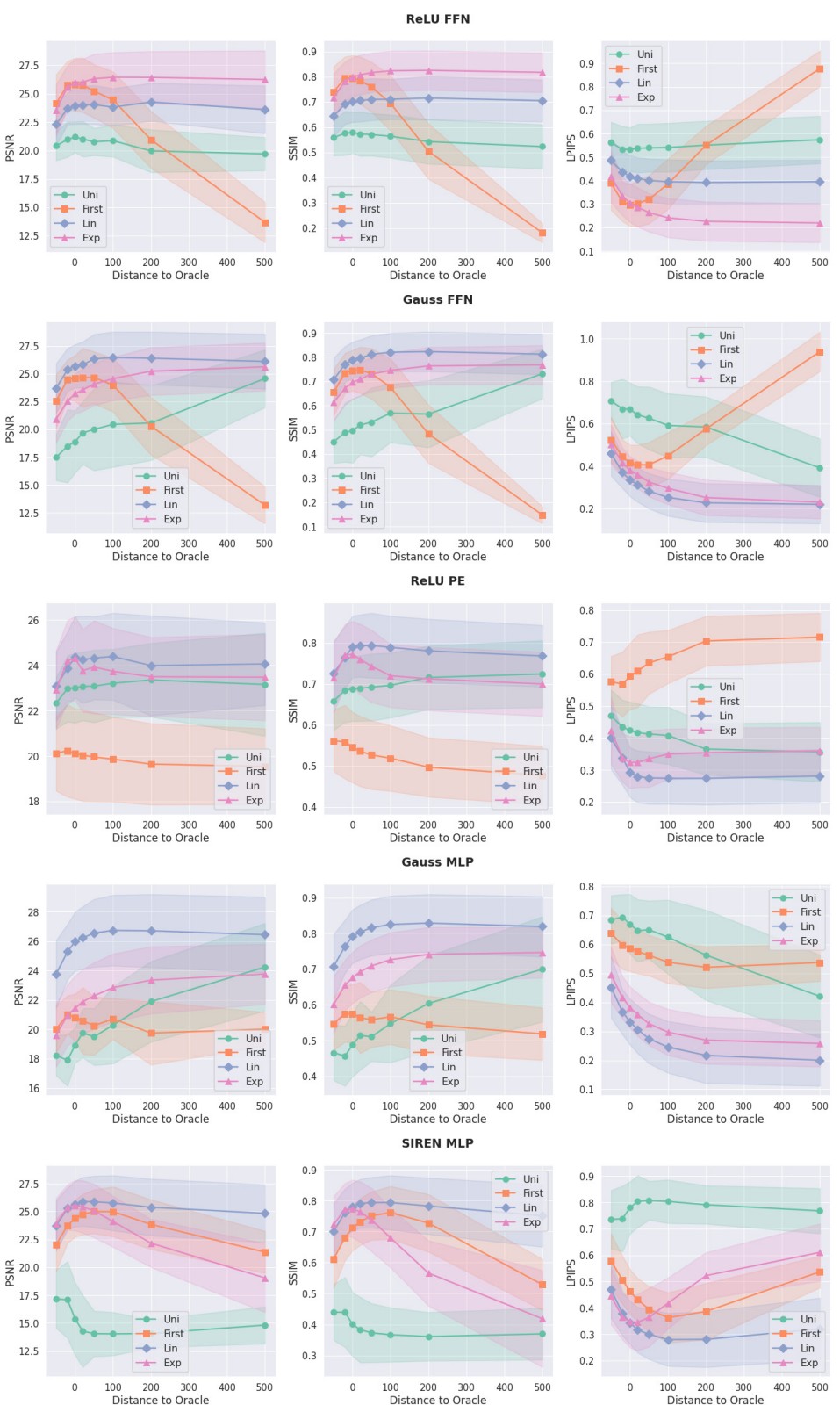

Figure 30: Budget Allocation for inpainting experiments for spectral normalization with $K_{min} = 0.25$

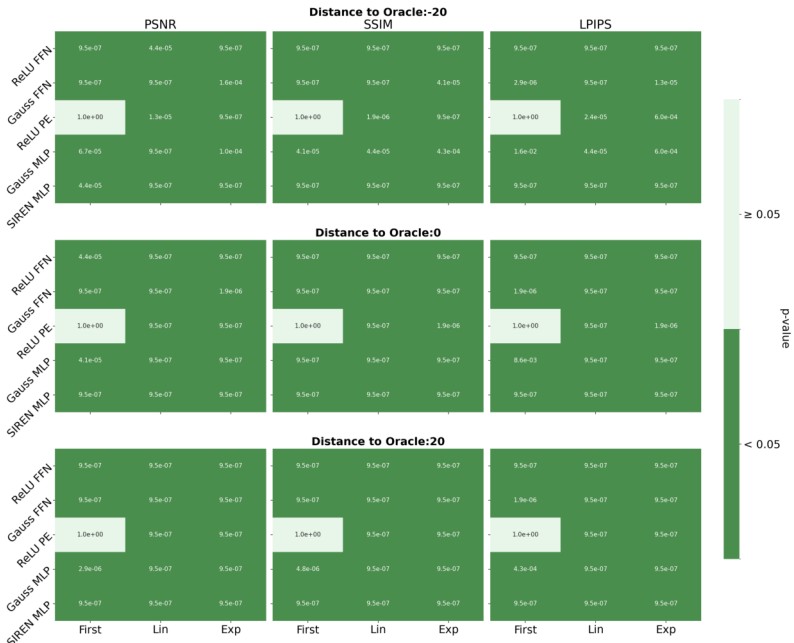

Figure 31: Heatmap for statistical significance test for inpainting experiment with spectral normalization with $K_{min} = 0.25$. Green indicates that non-uniform allocation is statistically better ($p < 0.05$) than uniform allocation. We observe that non-uniform strategies offer significant results in most cases across PSNR, SSIM, and LPIPS.

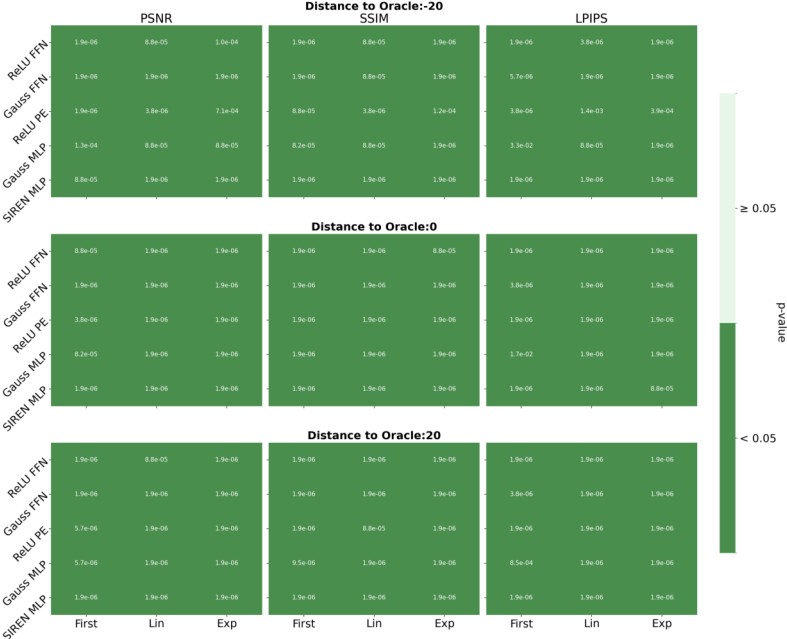

Figure 32: Heatmap for statistical significance test for inpainting experiment with spectral normalization with $K_{min} = 0.5$. Green indicates that non-uniform allocation is statistically better ($p < 0.05$) than uniform allocation. We observe that non-uniform strategies offer significant results in most cases across PSNR, SSIM, and LPIPS.

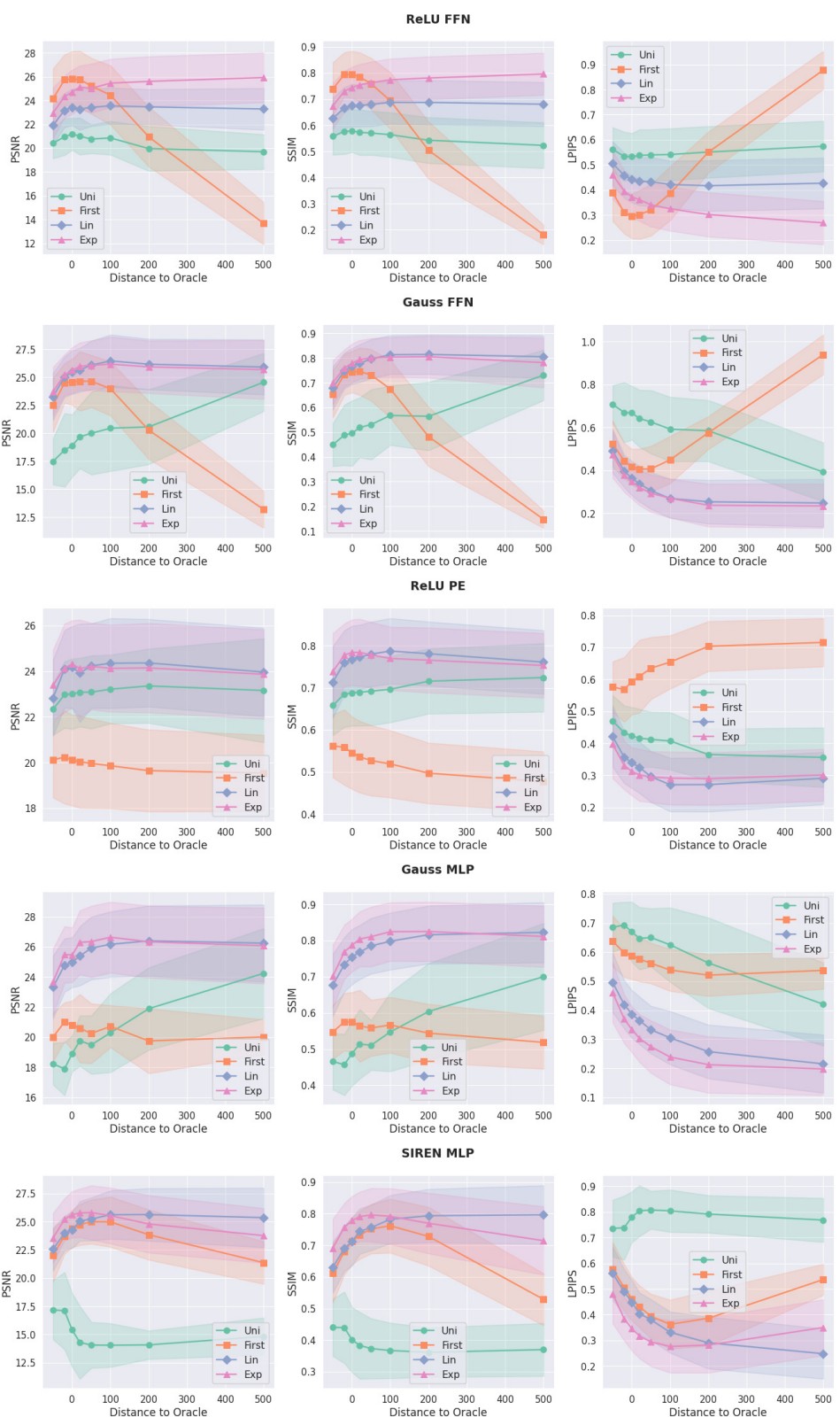

Figure 33: Budget Allocation for inpainting experiments for spectral normalization with $K_{min} = 0.5$

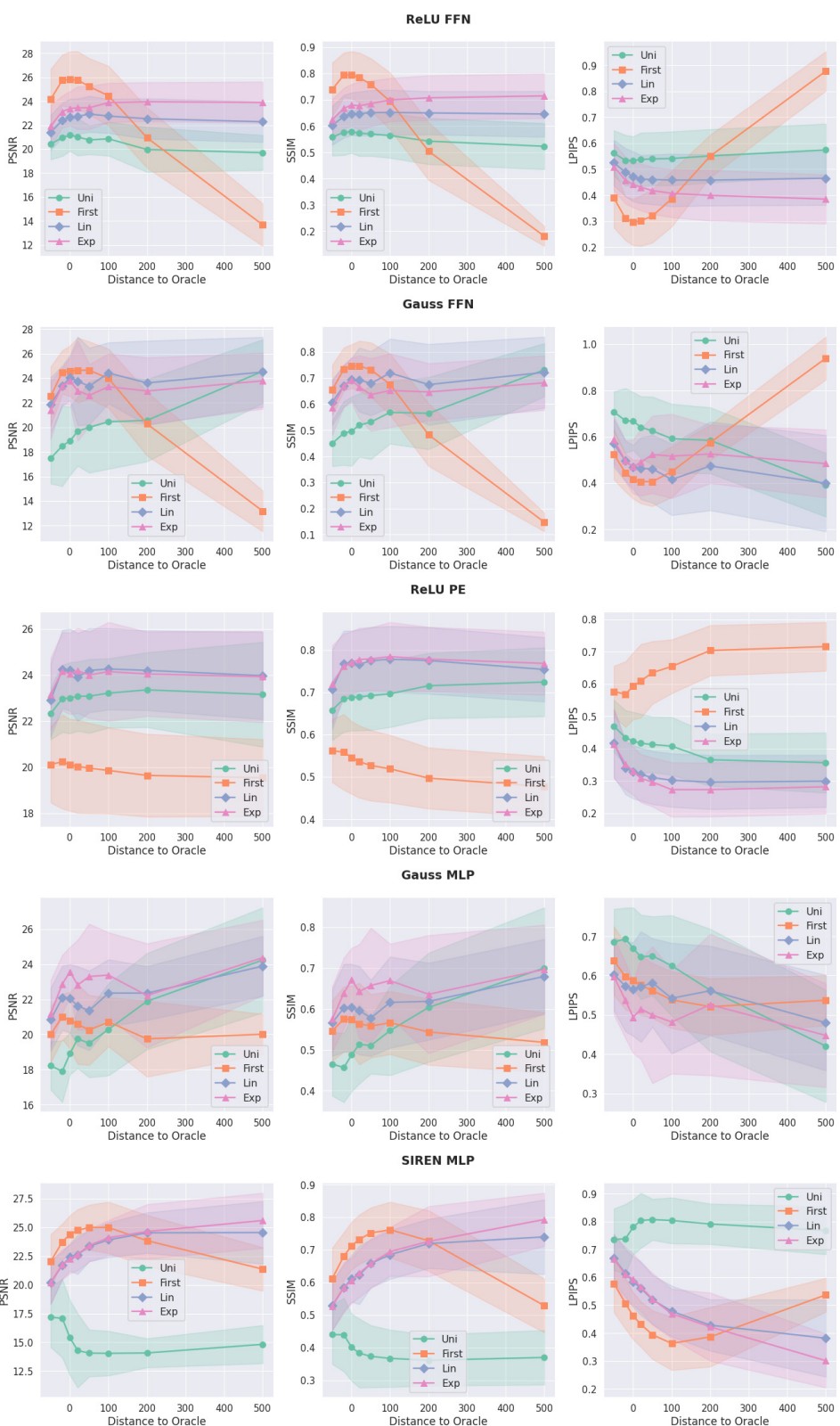

Figure 34: Budget Allocation for inpainting experiments for spectral normalization with $K_{min} = 1.0$

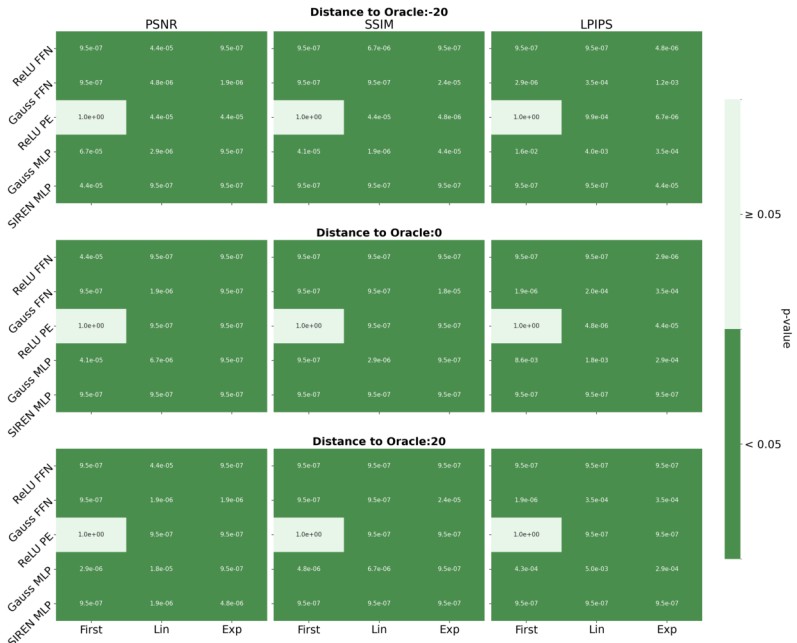

Figure 35: Heatmap for statistical significance test for inpainting experiment with spectral normalization with $K_{min} = 1.0$. Green indicates that non-uniform allocation is statistically better ($p < 0.05$) than uniform allocation. We observe that non-uniform strategies offer significant results in most cases across PSNR, SSIM, and LPIPS.

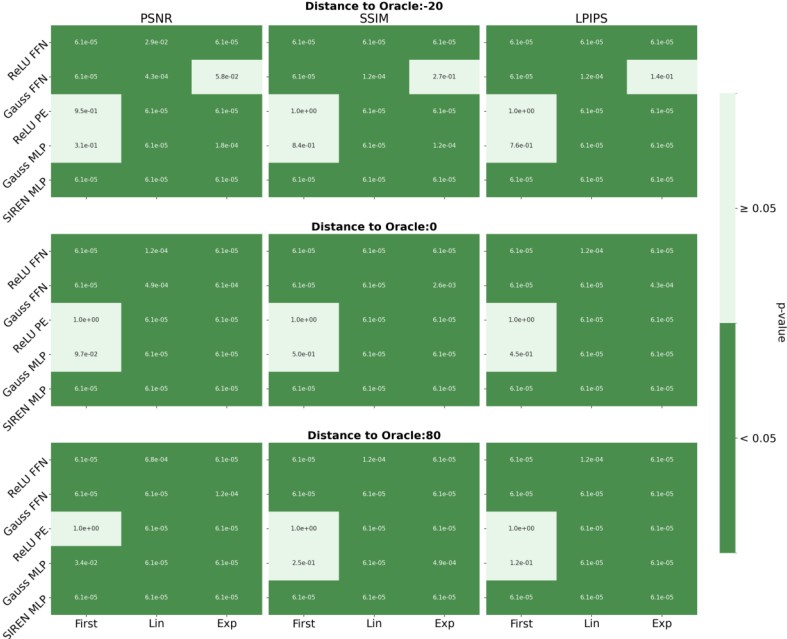

Figure 36: Heatmap for statistical significance test for super-resolution experiment with spectral normalization with $K_{min} = 0.25$. Green indicates that non-uniform allocation is statistically better ($p < 0.05$) than uniform allocation. We observe that non-uniform strategies offer significant results in most cases across PSNR, SSIM, and LPIPS.

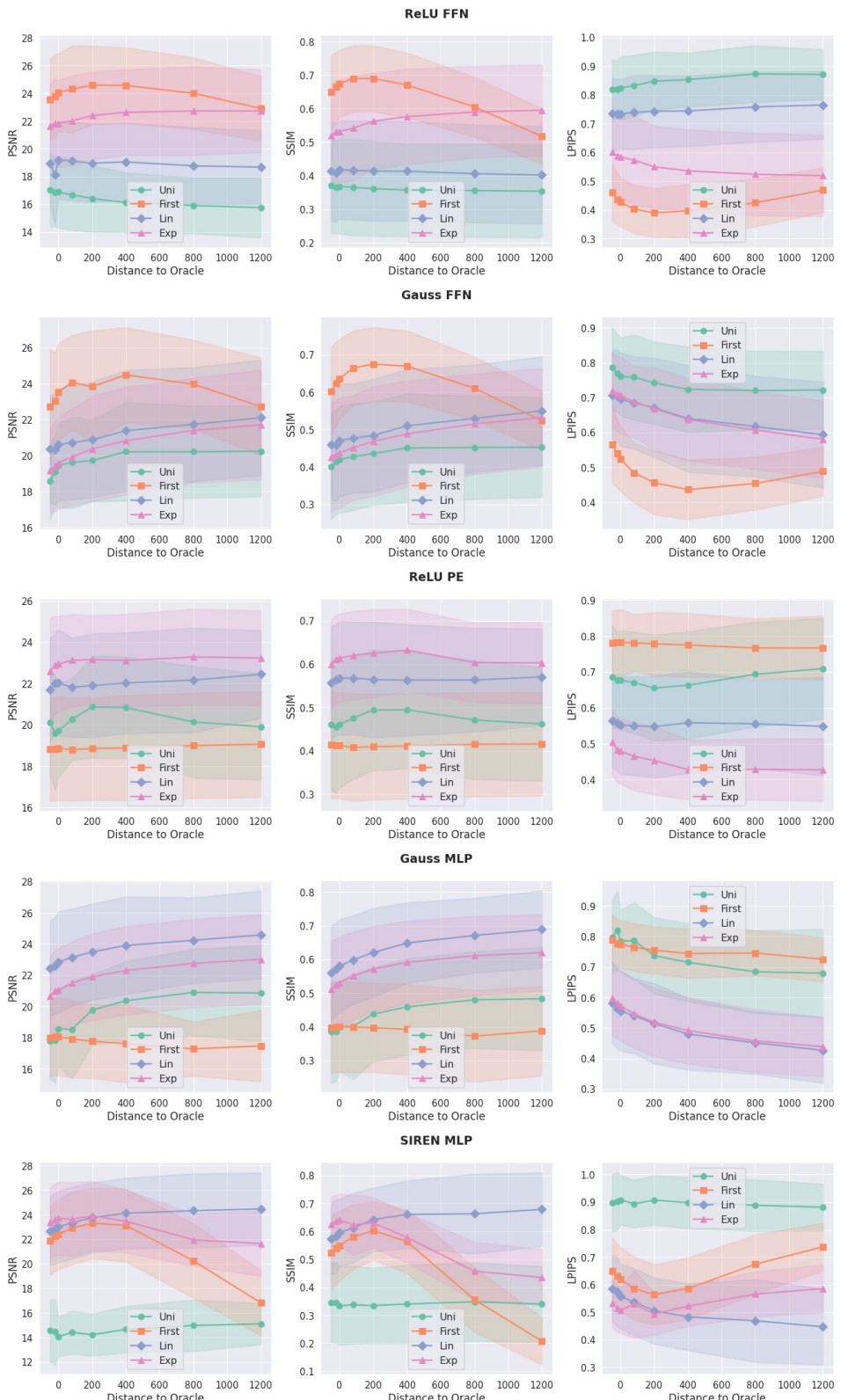

Figure 37: Budget Allocation for super-resolution experiments for spectral normalization with $K_{min} = 0.25$

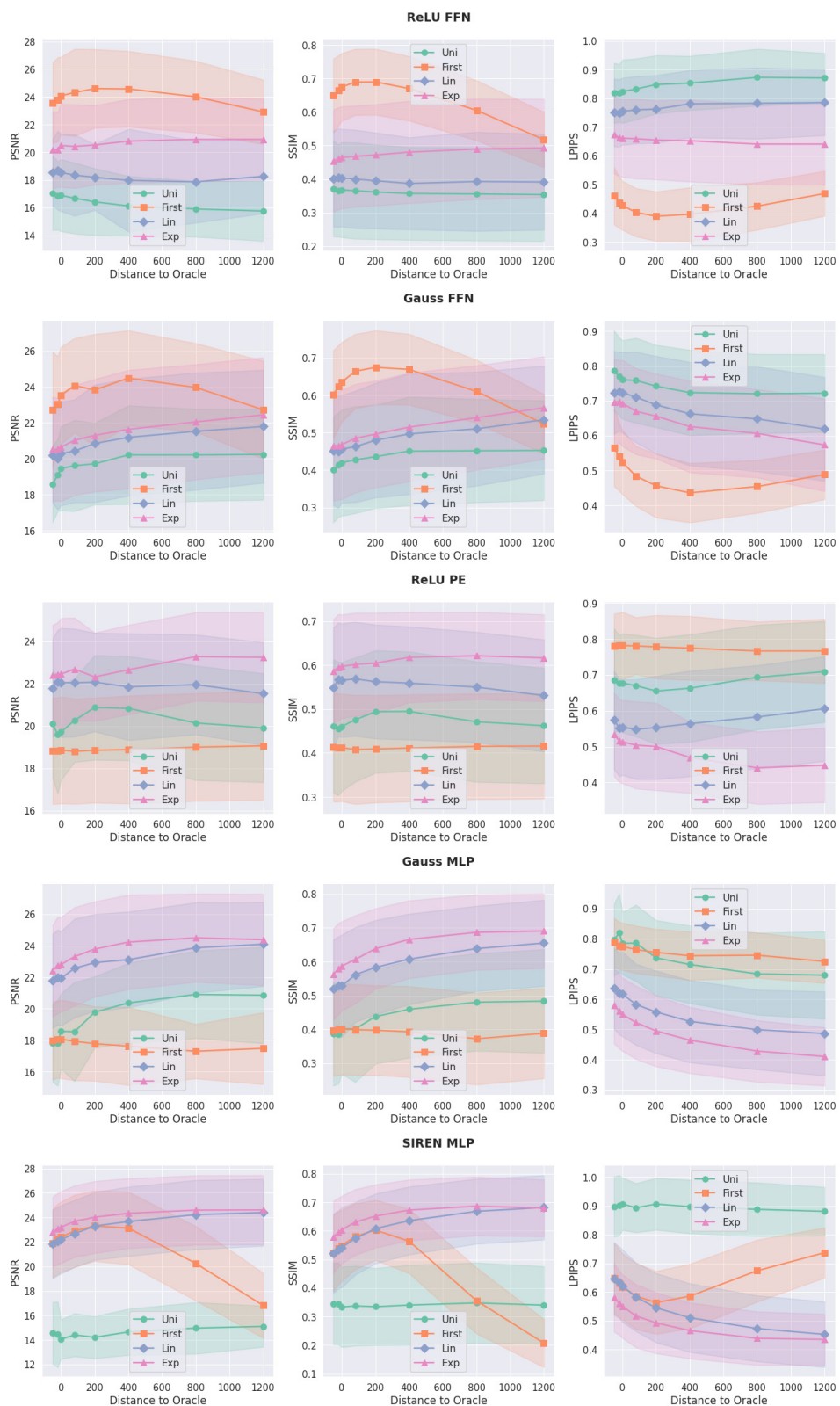

Figure 38: Budget Allocation for super-resolution experiments for spectral normalization with $K_{min} = 0.5$

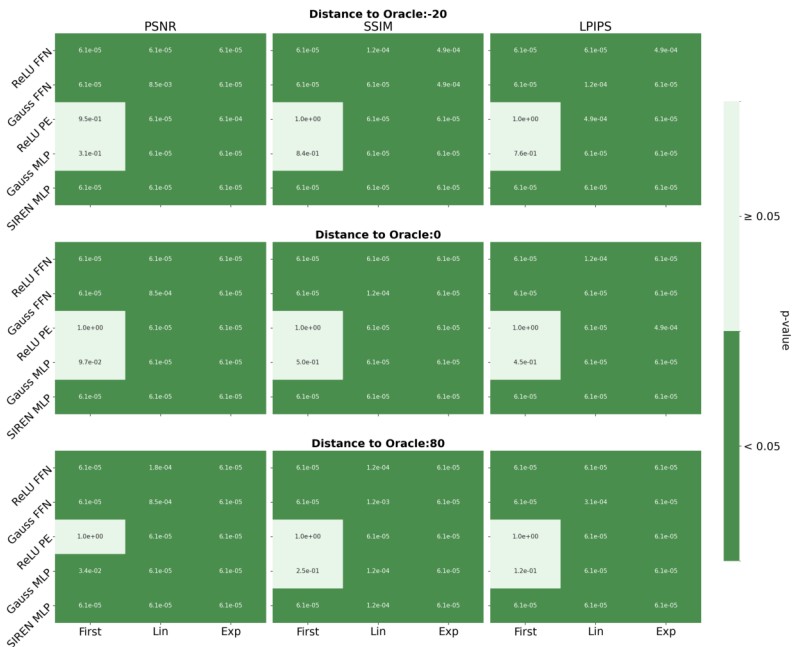

Figure 39: Heatmap for statistical significance test for super-resolution experiment with spectral normalization with $K_{min} = 0.5$. Green indicates that non-uniform allocation is statistically better ($p < 0.05$) than uniform allocation. We observe that non-uniform strategies offer significant results in most cases across PSNR, SSIM, and LPIPS.

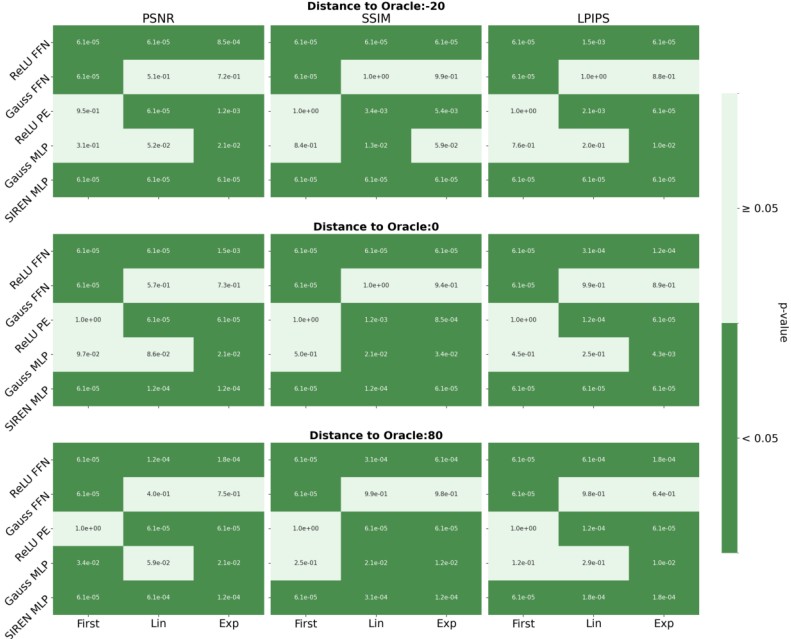

Figure 40: Heatmap for statistical significance test for super-resolution experiment with spectral normalization with $K_{min} = 1.0$. Green indicates that non-uniform allocation is statistically better ($p < 0.05$) than uniform allocation. We observe that non-uniform strategies offer significant results in most cases across PSNR, SSIM, and LPIPS.

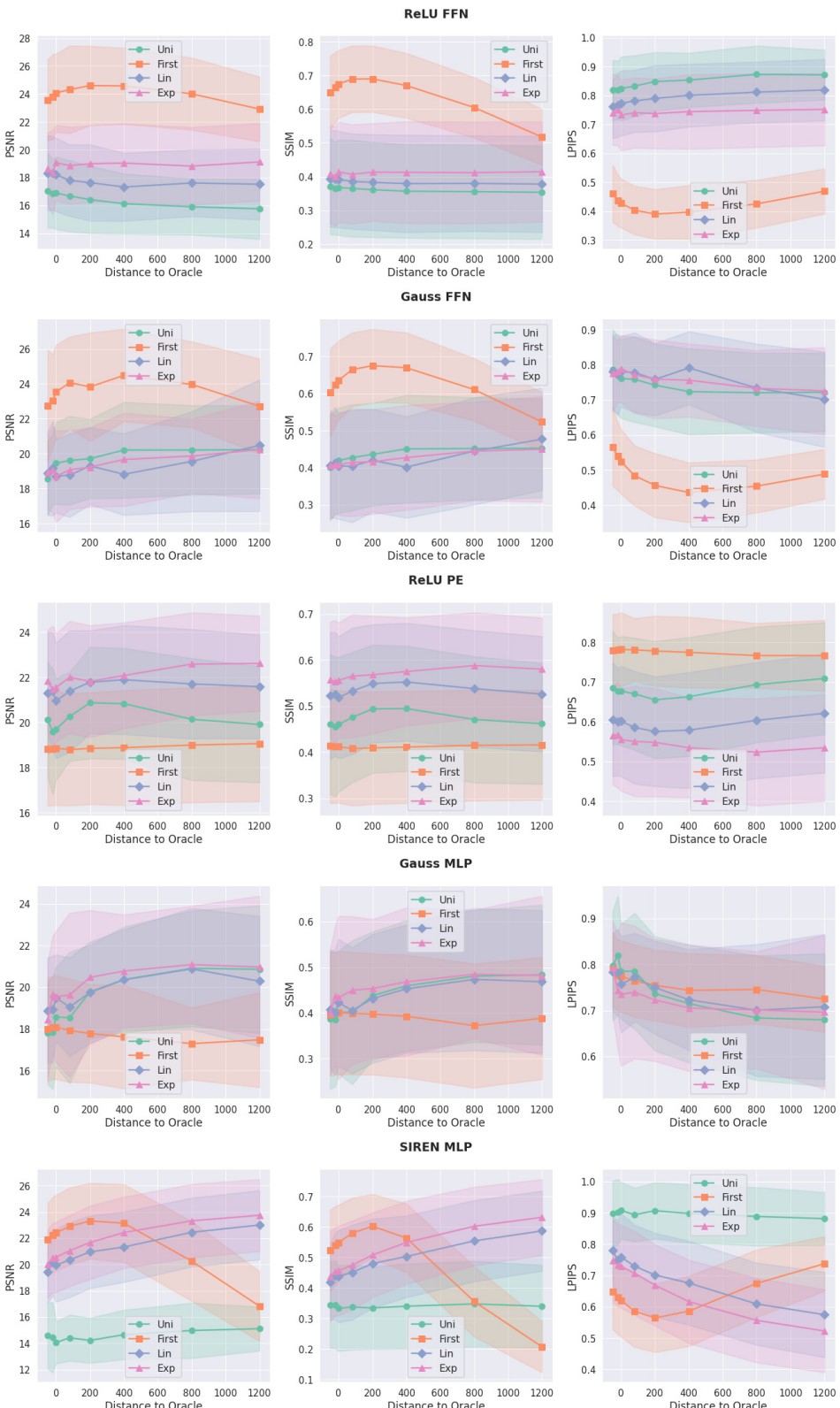

Figure 41: Budget Allocation for super-resolution experiments for spectral normalization with $K_{min} = 1.0$

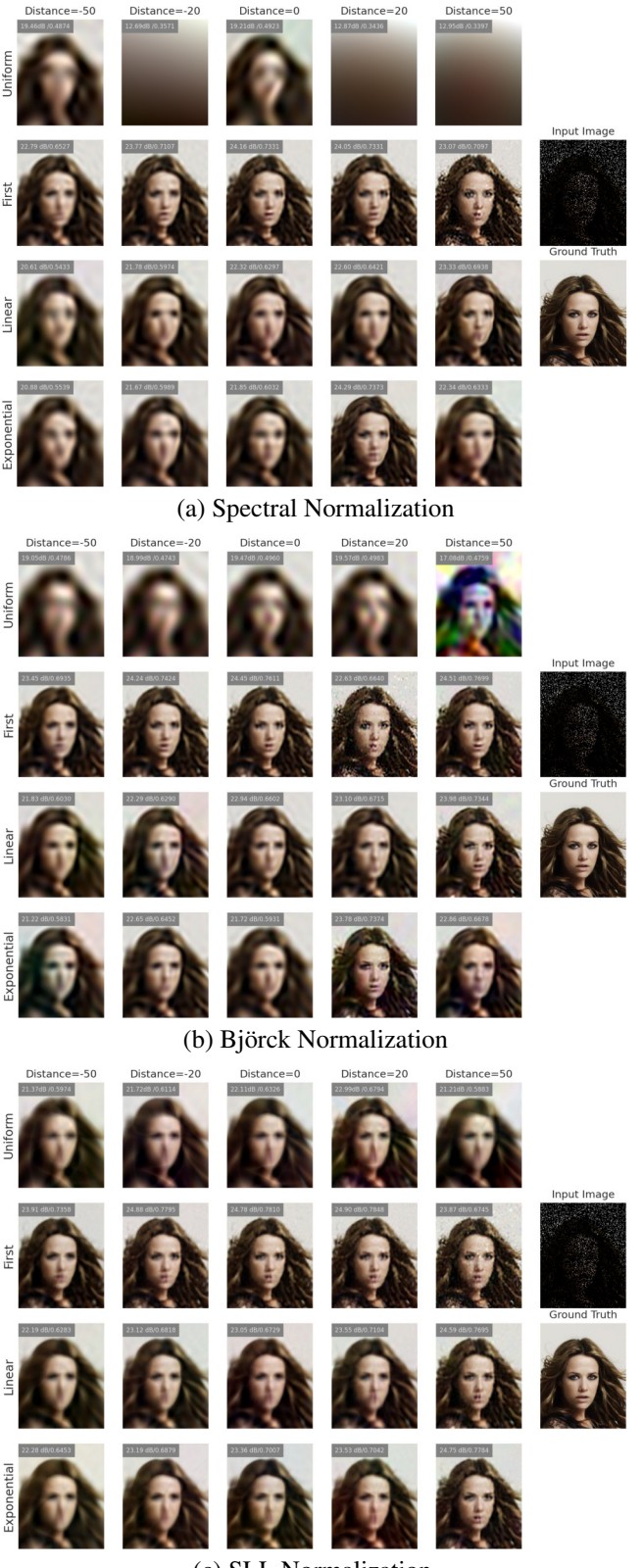

Figure 42: Qualitative examples from super-resolution experiments.

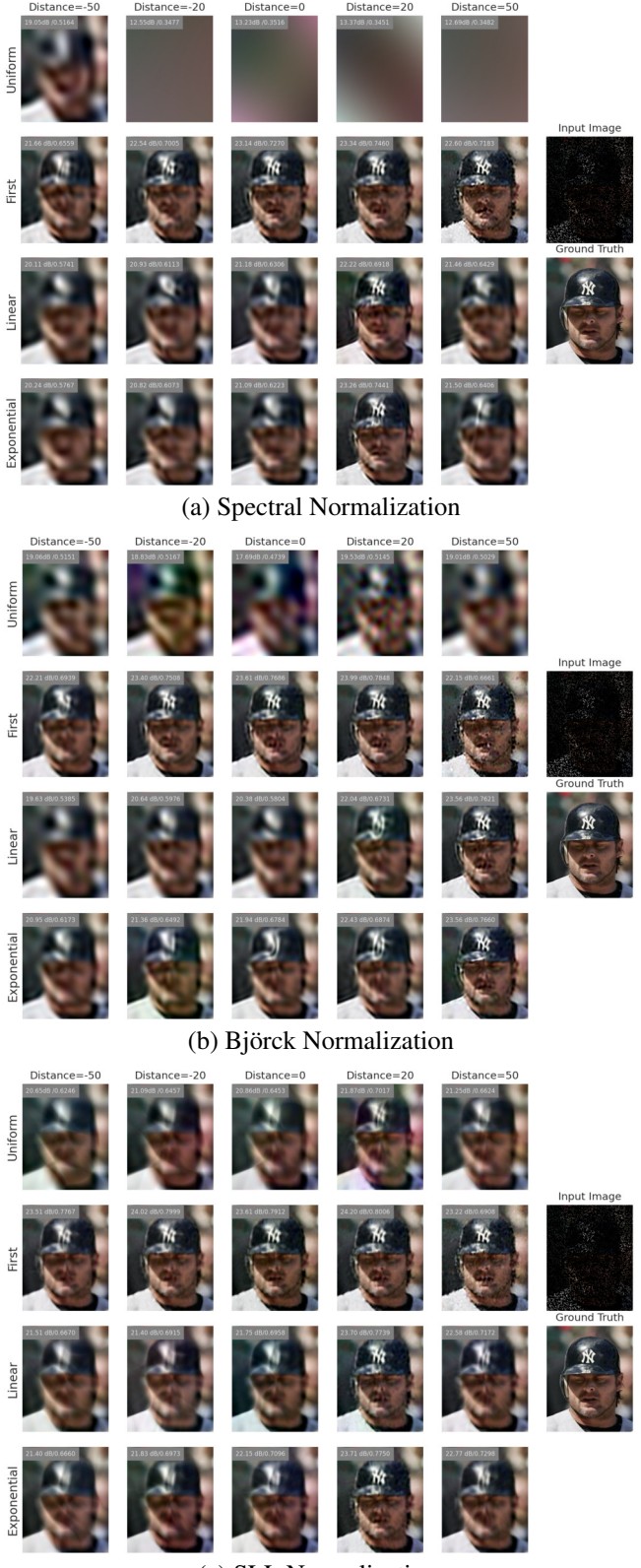

Figure 43: Qualitative examples from super-resolution experiments.

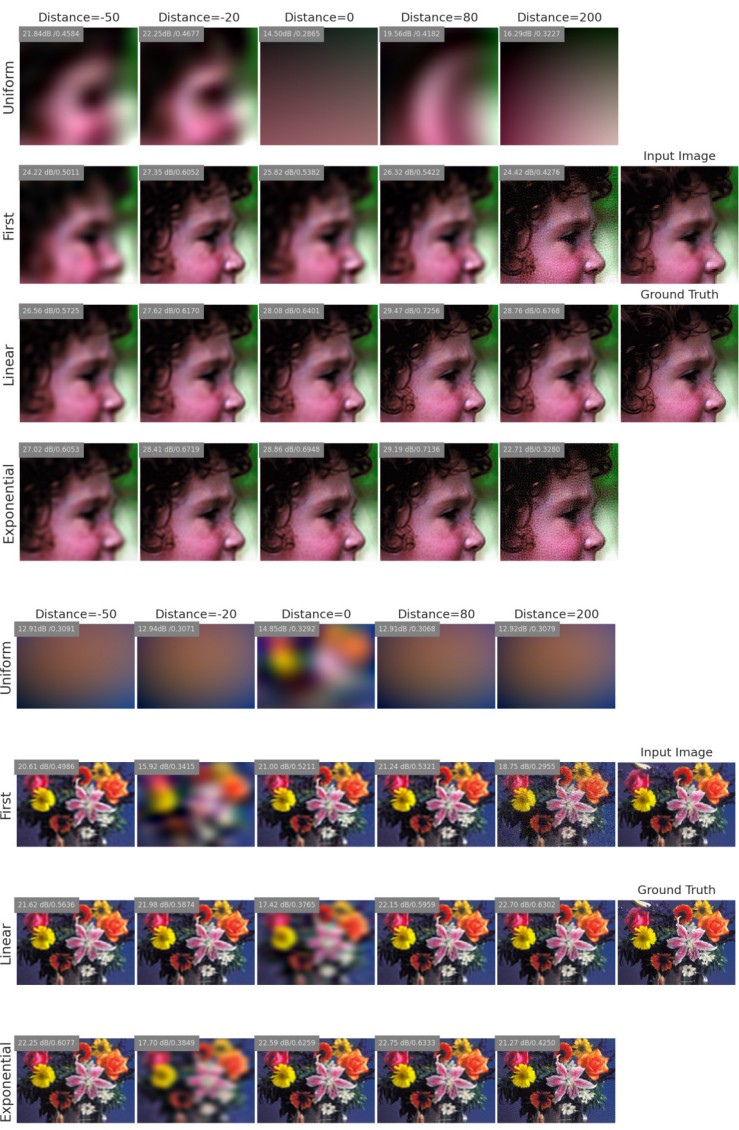

Figure 44: Qualitative examples from super-resolution experiments using spectral normalization.

