# OpenReview forum: "Beyond Uniformity: Regularizing Implicit Neural Representations through a Lipschitz Lens"
_ICLR.cc/2026/Conference — ICLR 2026 Poster_

### Official Review · Reviewer_ane9 · 2025-10-30

**Soundness:** 3
**Presentation:** 2
**Contribution:** 3
**Rating:** 6
**Confidence:** 3

**Summary:**

This paper revisits Lipschitz regularization in Implicit Neural Representations and introduces a K-Lipschitz budget framework that generalizes the traditional 1-Lipschitz constraint. The authors argue that uniform Lipschitz constraints (applied equally to all layers and components) are overly rigid and propose distributing a global Lipschitz budget non-uniformly across network components to balance smoothness and expressiveness. Experiments on signed distance fields (SDFs), medical image registration, and image inpainting demonstrate that flexible K-Lipschitz budgets improve stability, interpretability, and generalization.

**Strengths:**

- The idea of framing Lipschitz regularization as a budget allocation problem is fresh and theoretically well-motivated. It bridges a gap between rigid Lipschitz constraints and flexible, task-adaptive regularization schemes.
- The paper validates the theory across diverse INR applications—SDFs (geometry), deformable registration (medical imaging), and inpainting (vision)—showing generality.
- The Lipschitz lens provides a unifying view that complements NTK and Fourier analyses, contributing to a deeper understanding of implicit regularization in INRs.

**Weaknesses:**

- While the qualitative results are compelling, quantitative improvements (e.g., PSNR/SSIM/Folding Ratio) over strong baselines are small and sometimes marginal, with statistical significance not rigorously reported.
- I have some concerns regarding the oracle- and domain-driven estimates of K, but the estimation procedures are only sketched conceptually. Appendix references are insufficient for reproducing real-world tasks.
- Although the paper proposes an interesting conceptual framework (the Lipschitz-budget idea), most figures are purely experimental, focusing on results such as SDF reconstructions, lung deformation fields, and metric plots. The lungs in Fig.1 are quite confusing.

**Questions:**

- I have some questions about the fairness of comparison. The authors mention that all models are trained for a fixed number of epochs. However, since different architectures (e.g., SIREN vs. FFN vs. Gaussian-INR) and normalization methods may converge at different rates, a uniform epoch budget could introduce bias — some models might underfit or overfit relative to others. A convergence-based or early-stopping criterion would be a fairer comparison.
- What is the actual runtime or GPU memory overhead introduced by enforcing non-uniform Lipschitz constraints compared to the standard 1-Lipschitz setup?

---

> ### Author Response · Authors · 2025-11-26
> **Response to Reviewer ane9 - Part 1**
>
> Thank you for your insightful feedback. We answer all of your questions below.
>
> >While the qualitative results are compelling, quantitative improvements (e.g., PSNR/SSIM/Folding Ratio) over strong baselines are small and sometimes marginal, with statistical significance not rigorously reported.
>
> Thank you for the opportunity to clarify our results. While we have already reported confidence intervals in our plots in our initial submission, we were missing statistical tests. Following your feedback, we have now conducted statistical significance tests and report them in Appendix G4. In experiments with a budget K > 1 (registration, inpainting, and the newly added super-resolution experiment in Appendix B), the performance gain of non-uniform over uniform allocation in quantitative metrics is statistically significant.
>
> >I have some concerns regarding the oracle- and domain-driven estimates of K, but the estimation procedures are only sketched conceptually. Appendix references are insufficient for reproducing real-world tasks.
>
> We appreciate the opportunity to clarify details regarding the estimation of the budget K.  We have already provided the algorithm for the data-driven estimate in the appendix of our initial submission, along with an anonymized repository containing the code [Python implementation](https://anonymous.4open.science/r/iclr\_lip-88B1/sdf\_inpainting\_experiment/oracle.py). Regarding the domain-driven estimate, we contextualized the estimated value of K=2 using established physiological/medical literature, as described in Section 5.1.
>
> >Although the paper proposes an interesting conceptual framework (the Lipschitz-budget idea), most figures are purely experimental, focusing on results such as SDF reconstructions, lung deformation fields, and metric plots. The lungs in Fig.1 are quite confusing.
>
> Thank you for the feedback. Our intuition for Figure 1 was to (i) provide intuition for domain-driven Lipschitz estimates corresponding to actual, biological information on tissue stretch and (ii) to convey the budget distribution idea (uniform vs non-uniform). To better convey this message, we have added a more thorough explanation for this in the caption now.
>
> >I have some questions about the fairness of comparison. The authors mention that all models are trained for a fixed number of epochs. However, since different architectures (e.g., SIREN vs. FFN vs. Gaussian-INR) and normalization methods may converge at different rates, a uniform epoch budget could introduce bias — some models might underfit or overfit relative to others. A convergence-based or early-stopping criterion would be a fairer comparison.
>
> Thank you for this interesting question. We would like to clarify our setup.
>
> Firstly, while different architectures, such as SIREN, FFN, and Gaussian-INR, exhibit distinct training dynamics, they are typically trained for the same number of iterations in current INR benchmark papers \[1, 2\].
>
> Moreover, in our experiments, we do not aim to compare different architectures against each other. Instead, we only compare results **within the same architecture and normalization setting** to isolate the effect of the allocation strategy. For this purpose, we consider a fixed number of gradient steps to be the fairest evaluation protocol.
>
> \[1\] Essakine A, Cheng Y, Cheng CW, Zhang L, Deng Z, Zhu L, Schönlieb CB, Aviles-Rivero AI. *Where do we stand with implicit neural representations? A technical and performance survey.* TMLR
> \[2\] Kim N, Fridovich-Keil S. *Grids Often Outperform Implicit Neural Representations.* NeurIPS 2025

---

> ### Author Response · Authors · 2025-11-26
> **Response to Reviewer ane9 - Part 2**
>
> >What is the actual runtime or GPU memory overhead introduced by enforcing non-uniform Lipschitz constraints compared to the standard 1-Lipschitz setup?
>
> Thank you for the opportunity to clarify the computation cost. Please note that the choice of allocation strategy introduces negligible computational overhead compared to a uniform allocation strategy. The difference lies only in a one-time, static computation of the component-wise Lipschitz bounds (layer, embedding, and activation) during model initialization. This is either obtained in closed form or requires solving a computationally light one-dimensional root-finder equation in log-budget space using the [bisection method](https://anonymous.4open.science/r/iclr_lip-88B1/sdf_inpainting_experiment/lipschitz_budget_last.py). Regarding training/optimization, there is no additional computation/memory overhead when using a non-uniform Lipschitz INR.
>
> We additionally compare a Lipschitz INR against a non-Lipschitz INR below. For runtime computation, we use a vanilla 5-layer SIREN trained for 50 gradient steps and average the results over 10 trials.
>
> | Metric | SIREN (Mean ± Std) | LipSIREN (Mean ± Std) | Diff % |
> | :---- | :---- | :---- | :---- |
> | Training Time (s) | 0.523 ± 0.129 | 0.583 ± 0.038 | \+11.5% |
> | Peak Memory (MB) | 167.3 ± 0.0 | 167.8 ± 0.0 | \+0.3% |
>
> While the memory overhead is negligible (\+0.3%), spectral normalization (SN) introduces a slight increase (\+11.5%) in training time.
>
> Thank you for the great feedback, which has improved the paper’s presentation. Please let us know if our rebuttal has clarified your questions and comments. We would be happy to provide further clarification where necessary.

---

### Official Review · Reviewer_apk7 · 2025-10-30

**Soundness:** 3
**Presentation:** 4
**Contribution:** 3
**Rating:** 6
**Confidence:** 4

**Summary:**

This paper reframes Lipschitz regularization for Implicit Neural Representations (INRs) as a Lipschitz budget allocation problem. Rather than enforcing a strict 1-Lipschitz constraint, the authors treat the global Lipschitz constant K_B as a finite budget to be distributed non-uniformly across layers, activations, and embeddings. They derive layer-wise Lipschitz constants for common encodings, propose practical allocation strategies (uniform, linear, exponential, cosine), and demonstrate the approach on signed distance field reconstruction, deformable registration, and image inpainting. Empirically, non-uniform allocation improves the expressiveness-smoothness trade-off, and a data-driven oracle for estimating K_B yields interpretable, task-specific regularization.
The theoretical framing is sound: the derivations for per-component Lipschitz constants and the multiplicative composition rule are correct. The oracle-based Lipschitz budget estimation is mathematically consistent, though potentially sensitive to noise. Experiments are well designed and align with the hypotheses, though the product bound on Lipschitz constants is known to be loose, and the oracle lacks robustness analysis.
The paper is clearly written and well organized, with strong visual explanations and thorough appendices detailing hyperparameters, derivations, and code. Overall, the work unifies several regularization heuristics under a single “Lipschitz budget” framework. While the underlying tools are known, the reinterpretation and empirical synthesis are novel and practically valuable. The insights on how non-uniform allocation shapes expressivity and how Lipschitz analysis connects to NTK and Fourier perspectives provide novel intuition.

**Strengths:**

1. Conceptually elegant reframing of Lipschitz regularization as a quantitative resource-allocation problem.
2. Derivations for sinusoidal and random Fourier feature encodings connect spectral and geometric views.
3. Extensive appendices with reproducible setups (SDFs, registration, inpainting).
4. Empirical results consistent across modalities: tighter Lipschitz enforcement improves fidelity up to a task-dependent optimum.

**Weaknesses:**

1. The oracle estimator of K_B is a finite difference approximation, which is highly sensitive to discretization and noise. The paper lacks a rigorous analysis of this sensitivity
2. The product bound on Lipschitz constants is conservative; no tighter or data-dependent alternative has been explored.
3. The validation is on a small-scale dataset, e.g., Bunny SDFs, 256×256 inpainting, and a single lung registration pair.
4. No adaptive or learning mechanism for choosing K_B, for example, bi-level optimization
5. The method does not guarantee a diffeomorphism resulting in folds; a comparison with diffeomorphic baselines would have been helpful.

**Questions:**

1.	Could the authors provide empirical estimates of the actual Lipschitz constant of trained models and compare them to the theoretical budget?
2.	How sensitive is performance to noise or discretization in the oracle K_B estimation?
3.	Would a bilevel optimization approach that learns K_B automatically yield comparable results?
4.	For registration, have the authors compared with diffeomorphic baselines that explicitly constrain Jacobian determinants?

---

> ### Author Response · Authors · 2025-11-26
> **Response to Reviewer apk7 - Part 1**
>
> We would like to thank you for your insightful feedback and provide a detailed response below.
>
> >The product bound on Lipschitz constants is conservative; no tighter or data-dependent alternative has been explored.
> >Could the authors provide empirical estimates of the actual Lipschitz constant of trained models and compare them to the theoretical budget?
>
> Thank you for the great feedback. In general, measuring tight Lipschitz bounds is an active field of research and challenging, especially for wide and deep MLPs. This is further complicated by INRs’ various activation functions, which limit the applicability and accuracy of state-of-the-art SDP-based frameworks typically used for ReLU and other 1-Lipschitz activations, but not for sinusoidal/Gaussian activation functions.
>
> Consequently, we utilize an empirical estimation strategy introduced in \[1\] instead and include the empirical measurements of the network Lipschitz constant in Figure 3\. For a brief summary of our results, please refer to our general response. In brief, these results reaffirm that Lipschitz INRs using Bjoerck or SLL-based normalization closely approach the upper Lipschitz bound (e.g., a measured network Lipschitz constant of approximately 0.95 at K=1 for shapes).
>
> \[1\] Wang R, Manchester I. Direct parameterization of lipschitz-bounded deep networks. ICML’23
>
> >The validation is on a small-scale dataset, e.g., Bunny SDFs, 256×256 inpainting, and a single lung registration pair.
>
> We appreciate the opportunity to clarify our experimental setup. We would like to note that we use similar sample sizes to those in the current INR benchmark papers \[1,2\]. We employ 10 shapes, 25 images for inpainting (218x178 pixels), and the entire DIRLAB dataset (10 patients) for registration.
>
> \[1\] Kim N, Fridovich-Keil S. Grids Often Outperform Implicit Neural Representations. arXiv preprint arXiv:2506.11139. 2025 Jun 10\.
> \[2\] Essakine A, Cheng Y, Cheng CW, Zhang L, Deng Z, Zhu L, Schönlieb CB, Aviles-Rivero AI. Where do we stand with implicit neural representations? a technical and performance survey. arXiv preprint arXiv:2411.03688. 2024 Nov 6\.
>
> >The oracle estimator of K\_B is a finite difference approximation, which is highly sensitive to discretization and noise. The paper lacks a rigorous analysis of this sensitivity
> >How sensitive is performance to noise or discretization in the oracle K\_B estimation?
>
> Thank you for this valuable suggestion. To address this, we have performed a sensitivity analysis for the oracle estimate on photon noise, which is now detailed in Appendix C2. Adopting the noise simulation of \[1\], we compare gradient distributions under high-noise conditions against our original estimates. Our results indicate that although noise impacts the tail of the distribution, the critical percentiles (99.9% and 99.99%) remain comparable to the original estimate. This makes a strong case for our oracle estimate and further demonstrates its utility in the presence of noise.
>
> Regarding discretization, we would like to clarify that the grid resolution fundamentally alters the
> Lipschitz constant of the signal to be modeled by the INR, since the input domain is always scaled to a range of \[-1,1\]. This necessitates that our oracle is determined at the desired resolution. Thus, “discretization” is an explicitly modeled feature of our oracle estimate.
>
> \[1\] Saragadam V, LeJeune D, Tan J, Balakrishnan G, Veeraraghavan A, Baraniuk RG. Wire: Wavelet implicit neural representations. In Proceedings of the IEEE/CVF Conference on Computer Vision and Pattern Recognition 2023 (pp. 18507-18516).

---

> ### Author Response · Authors · 2025-11-26
> **Response to Reviewer apk7 - Part 2**
>
> >For registration, have the authors compared with diffeomorphic baselines that explicitly constrain Jacobian determinants?
>
> Following your suggestion, we evaluated two additional baselines to benchmark the trade-off between expressiveness (TRE) and transformation smoothness (folding ratio):
>
> - A vanilla Siren with Jacobian regularization explicitly targeting transformation smoothness (i.e., a low folding ratio).
> - NODEO\[1\]: A learning-based diffeomorphic registration algorithm designed to strictly avoid folding.
>
> We report the baseline experiment in Appendix F, and briefly comment on the results:
>
> While Lipschitz INRs are *not* specifically designed to avoid folding, but rather to model anatomically plausible transformations by constraining the maximum stretch, they remain particularly effective in our experiments and strike a good balance between expressiveness (TRE) and smoothness (folding ratio). For the folding ratio, they outperform SIREN, with Jacobian regularization, in 8 out of 10 cases. Importantly, Lipschitz SIRENs remain robust across all cases and do not fluctuate as strongly in performance as SIREN or NODEO.
> NODEO, as expected, yields the state-of-the-art folding ratio, but results in higher target registration error, given that its transformations are strictly constrained, posing a challenge to the learning of the transformation.
>
> \[1\] Wu Y, Jiahao TZ, Wang J, Yushkevich PA, Hsieh MA, Gee JC. Nodeo: A neural ordinary differential equation based optimization framework for deformable image registration. InProceedings of the IEEE/CVF conference on computer vision and pattern recognition 2022 (pp. 20804-20813).
>
> >Would a bilevel optimization approach that learns K\_B automatically yield comparable results?
>
> Thank you very much for this interesting idea, which has been similarly raised by reviewr m2hg.
>
> While simultaneously estimating K\_B during training may not be applicable in cases where the ideal Lipschitz budget is known (e.g., K=1 for shapes), this idea is relevant for inverse problems, where reference data or domain knowledge are not available.
>
> However, learning K\_B within the context of the training objective of INRs may encourage the network to raise K\_B, approaching an under-constarined or unconstrained scenario, as the INR’s loss function typically encourages overfitting. To counter this, one needs to devise a strategy within the typical INR training scheme to validate the K\_B and regularize against the network, increasing the budget. Lastly, making K\_B learnable may also introduce complexity and potentially destabilize the training dynamics. Nevertheless, we find this to be a fascinating line of research, but it falls outside the scope of the current work and is relevant in the context of future work.
>
> We appreciate your great feedback, which led to a significant improvement in the paper's presentation. Please let us know whether our rebuttal effectively addresses all your questions and comments. We are ready to provide any additional clarification that may be needed.

---

### Official Review · Reviewer_m2hg · 2025-10-31

**Soundness:** 3
**Presentation:** 3
**Contribution:** 2
**Rating:** 6
**Confidence:** 4

**Summary:**

This paper focuses on Lipschitz-constrained implicit neural representations (INRs). The authors propose to estimate a Lipschitz bound K using domain knowledge and explore different ways to distribute K across the components of the underlying neural network. They validate their framework on several tasks, including learning signed distance functions for object representation, image registration, and image inpainting. Overall, the paper builds on relatively simple and nice ideas and presents them clearly. The results are consistent with known trade-offs in Lipschitz-constrained networks.

**Strengths:**

Overall, the paper is easy to follow. There are extensive numerical experiments on different tasks that investigate the effect of different components. The integration of Lipschitz constraints into INRs seems like a nice line of research to explore.

**Weaknesses:**

Some parts could be improved (see questions).

**Questions:**

Could you include a few lines on how one should proceed with estimating the K bound? I know it is mentioned for each task, but I want a more general paragraph that tells what domain knowledge to look for and how to process estimating K when facing a new task. Could it be embedded into the training or form an optimization scheme to estimate it?

Could you give an intuition why the budget distribution is not that effective in the SDF example, but it makes a difference for the image inpainting task?

In Figure 2, why is the second bunny in row one different from the first of row two? Aren't they both with Bjoerick and ReLU?

In the description of Figure 4 is written "the Lipschitz-regularized model exhibits significantly reduced folding artifacts while maintaining comparable TRE and the anatomical plausibility of the deformation"; however, in the Figure, there is no indication of Lipschitz / non-Lipschitz model, which confuses me. Could you clarify?

Could you include visual examples (images) for the image inpainting?

I suggest moving the citations in the conclusion to other parts and trying to summarize your own work there.

I suggest more visual results (for the images) to help understand the effect of the methods.

---

> ### Author Response · Authors · 2025-11-26
> **Response to Reviewer m2hg**
>
> We thank you for the insightful review and provide answers to each of your questions below.
>
> >Could you include a few lines on how one should proceed with estimating the K bound? I know it is mentioned for each task, but I want a more general paragraph that tells what domain knowledge to look for and how to process estimating K when facing a new task. Could it be embedded into the training or form an optimization scheme to estimate it?
>
> This is an excellent question. Based on your feedback, we have decided to add a dedicated section to the main manuscript (Sec. 6\) to provide practitioners with a detailed guide on selecting and distributing budgets for new applications and problems. We also show its applicability in a new experiment on single-image super-resolution in Appendix B.
>
> We address your interesting question regarding the optimization of the budget K\_B, similarly raised by apk7.
>
> While simultaneously estimating K\_B during training may not be applicable in cases where the ideal Lipschitz budget is known (e.g., K=1 for shapes), this idea is relevant for inverse problems, where reference data or domain knowledge are not available.
>
> However, learning K\_B within the context of the training objective of INRs may encourage the network to raise K\_B, approaching an under-constarined or unconstrained scenario, as the INR’s loss function typically encourages overfitting. To counter this, one needs to devise a strategy within the typical INR training scheme to validate the K\_B and regularize against the network, increasing the budget. Lastly, making K\_B learnable may also introduce complexity and potentially destabilize the training dynamics. Nevertheless, we find this to be a fascinating line of research, but it falls outside the scope of the current work and is relevant in the context of future work.
>
> >Could you give an intuition why the budget distribution is not that effective in the SDF example, but it makes a difference for the image inpainting task?
>
> We believe this difference comes from the fundamental nature of the signals:
>
> Since SDFs should represent the shortest, orthogonal distance to a shape’s boundary, the gradient magnitude is required to be ∥∇f∥ \= 1 almost everywhere. This means the signal's Lipschitz objective is identical nearly everywhere. As the problem is spatially homogeneous, our intuition suggests that a uniform budget is ideal, i.e., there is limited variability in the desired signal properties.
>
> In contrast, natural images contain a mix of smooth regions and sharp edges, resulting in high spatial variability of local Lipschitzness. To represent this large dynamic range, we believe it is beneficial for the early layers to have a high level of expressiveness, enabling them to model a diverse set of features. Hence, we observed a higher benefit of non-uniform allocation strategies in the natural image tasks.
>
> >In Figure 2, why is the second bunny in row one different from the first of row two? Aren't they both with Bjoerick and ReLU?
>
> Thank you for bringing this to our attention \- these should indeed be identical. This is an error on our part, resulting from a wrong path in our script for creating the visualization. We have updated the figure accordingly.
>
> > In the description of Figure 4 is written "the Lipschitz-regularized model exhibits significantly reduced folding artifacts while maintaining comparable TRE and the anatomical plausibility of the deformation"; however, in the Figure, there is no indication of Lipschitz / non-Lipschitz model, which confuses me. Could you clarify?
> > Could you include visual examples (images) for the image inpainting?
> > I suggest more visual results (for the images) to help understand the effect of the methods.
>
> Referencing a non-Lipschitz model in the context of the uniform/non-uniform allocation experiments was indeed confusing, as the plot does not compare results to a non-Lipschitz baseline, but visualizes exemplary results of a registration experiment.
>
> To substantiate our initial claim, we have now added a Jacobian-regularized SIREN as a baseline in our experiments (c.f. App. F), and have revised the figure caption.
>
> Following your valuable suggestion to include visual examples for inpainting, we have decided to move the visualization of the registration to the Appendix, leaving us space to include a comparison for the inpainting task. Moreover, we have also added visualizations for various (other) architectures in the Appendix.
>
> >I suggest moving the citations in the conclusion to other parts and trying to summarize your own work there.
>
> Following your suggestion, we now discuss the cited papers in related work and have expanded the conclusion to discuss our framework.
>
> We thank you for the great feedback, which has greatly improved the paper’s presentation.
> Please let us know if our rebuttal has adequately addressed your questions and comments.
> We would be happy to provide further clarification if necessary.

---

### Official Review · Reviewer_aw4e · 2025-10-31

**Soundness:** 3
**Presentation:** 3
**Contribution:** 3
**Rating:** 4
**Confidence:** 3

**Summary:**

This paper proposes a K-Lipschitz budget framework for implicit neural representations, deriving task-specific budgets and exploring non-uniform allocation strategies across network components. The work provides theoretical foundations and demonstrates applications in deformable registration, signed distance fields, and inpainting.

**Strengths:**

Writing is clear.

**Weaknesses:**

* The paper claims uniform 1-Lipschitz allocation is suboptimal but Figure 3 shows minimal differences between allocation strategies at K=1. Why is non-uniform allocation necessary if uniform allocation of a different budget K could achieve the same effect?

* Results conflate budget value K, allocation strategy, and
  normalization method, which factor actually drives performance?

* The inpainting oracle estimates Lipschitz constant from finite differences on the discrete image. Why should a continuous neural representation mimic the Lipschitz constant of its discrete sampling?

* The paper positions Lipschitz constraints as addressing "lack of implicit regularization" but never compares to actual implicit regularization from optimization dynamics or simply training longer. Does explicit Lipschitz regularization outperform other forms of regularization or architectural constraints?

* Only 5 hand-designed parametric schedules are explored with minimal
  justification. Given that optimal allocation appears task-dependent.
  The paper promises "actionable principles" but provides no mechanism
  for deriving allocations for new tasks.

* Experiments use small samples without confidence intervals or significance testing. Claims like "significantly sharper reconstruction" rely on visual inspection despite high variance in plots, making it difficult to assess whether observed differences are meaningful.

**Questions:**

See above

---

> ### Author Response · Authors · 2025-11-26
> **Response to Reviewer aw4e - Part 1**
>
> We appreciate your thorough review and provide answers to your questions below.
>
> > The paper claims uniform 1-Lipschitz allocation is suboptimal but Figure 3 shows minimal differences between allocation strategies at K=1. Why is non-uniform allocation necessary if uniform allocation of a different budget K could achieve the same effect?
>
> Thank you very much for this interesting question.
>
> 1) Necessity of non-uniform allocation strategies in shape experiments:
> We pose the question of whether uniform allocation is optimal as a research question rather than a claim in the Introduction. We explore non-uniform budgets for shapes as part of this research question. As stated in our initial submission (Sec. 4.2), for shapes, non-uniform allocation is not crucial, which aligns with your observation and motivates us to examine different tasks with higher values of K, where we observe clear advantages of non-uniform allocation (Sec 5.1 and Sec 5.2).
>
> 2) Using a uniform allocation strategy with a different budget K:
> This is an interesting idea. For signed distance functions, the ideal theoretical setting is a 1-Lipschitz constraint. One might therefore ask: *Do Lipschitz INRs actually approach the imposed budget? And can the Lipschitz budget be safely relaxed to (K>1) if its effective usage is below 1, thereby allowing more expressiveness?* While we did not quantify this in the original manuscript, the revised version now includes empirical Lipschitz estimates (c.f. revised Fig. 2), demonstrating that Lipschitz-constrained INRs utilize the budget efficiently, typically approaching it closely (up to 0.96 in Fig. 2). In contrast, non-Lipschitz INRs substantially overshoot it (Measured Lipschitz \= 2.7). Consequently, using a higher uniform budget does not reproduce the same effect: the desired 1-Lipschitz property is no longer guaranteed, which can introduce undesirable artifacts and problematic behavior in downstream procedures such as marching cubes, consistent with the observations in \[1\].
>
> We thus recommend using tighter normalization strategies, such as SLL, that efficiently utilize the budget at K=1, rather than increasing the budget to K>1 and losing the guarantees of a desirable SDF \[1\].
>
> >Results conflate budget value K, allocation strategy, and normalization method, which factor actually drives performance?
>
> We took great effort to clearly disentangle the effects of the different influencing factors on performance, in particular, budget value K and allocation strategy, as we generally only varied one and kept the other constant. We briefly summarize this again in the context of our paper:
>
> Shapes:
> - Exp1 (a): We set K=1 and set the activation function to ReLU, and vary the normalization
> - Exp1 (b): We set K=1 and set the normalization to Bjoerck, and vary the activation function
> - Exp2: We set K=1, and we compare uniform vs non-uniform within the same architecture/normalization.
>
> Registration:
> - We set K \= 2 and compare uniform versus non-uniform within the same architecture/normalization.
>
> Inpainting:
> - We estimate the budget K.
> - We run the same experimental setup, i.e., comparison of the uniform/non-uniform allocation strategies, within the same architecture/normalization and distances to the oracle (e.g., \-50, 0, 50).
>
> To answer your question about which *factor* drives performance, we summarize our observations as follows:
>
> In practice, the performance of an INR under a given Lipschitz regularization is shaped by a combination of factors rather than any single one: (1) the choice of activation function (e.g., SIREN, Gaussian), (2) how amenable the architecture is to different normalization methods, and (3) the chosen budget allocation strategy.
>
> 1) For shapes, as discussed in Section 4.2 of our paper, we observe a trend where the architecture (normalization \+ activation function) is a significant driver of reconstruction quality. Here, the budget allocation strategies are not the primary driver.
> 2) For the registration experiments, where we compare budget allocation strategies within the same architecture/setup, we observe K\_min (i.e., the minimum Lipschitz constant of a network component) as a significant driver of performance, where lower K\_min (relevant for non-uniform strategies) leads to better TRE.
> 3) For the inpainting, we observe differences in both architecture and allocation strategy. (i) For the same architecture and budget, we see substantial improvements in non-uniform allocation strategies over uniform, making the choice of allocation a great performance driver. (ii) For the same budget K and allocation strategy, we also see normalization as a relevant factor in driving the performance, where certain normalizations, such as SLL, yield higher quality. (iii) Lastly, setting the global budget K to a meaningful estimate, as per our oracle, is critical, as observed in our experiment. For the same allocation and architecture, an ill-set budget yields sub-optimal performance.

---

> ### Author Response · Authors · 2025-11-26
> **Response to Reviewer aw4e - Part 2**
>
> >The inpainting oracle estimates Lipschitz constant from finite differences on the discrete image. Why should a continuous neural representation mimic the Lipschitz constant of its discrete sampling?
>
> Thank you very much for this great question, and would like to provide our intuition on this.
>
> So far, practitioners lack guidance for choosing an appropriate Lipschitz budget (K) for INRs. Since we only have discrete observations and INRs train on them, we view the discrete oracle estimate as an efficient, viable solution. Although this estimate does not equal the (unknown) Lipschitz constant of the true continuous function, our experiments indicate that it provides a sufficiently accurate and practically useful approximation. Moreover, from a signal-theoretic perspective, under standard assumptions such as bandlimitedness and sampling above the Nyquist rate, the discrete oracle yields a meaningful and well-justified estimate of the underlying smoothness, making it a reasonable target for the continuous INR to mimic.
>
> >The paper positions Lipschitz constraints as addressing "lack of implicit regularization" but never compares to actual implicit regularization from optimization dynamics or simply training longer. Does explicit Lipschitz regularization outperform other forms of regularization or architectural constraints?
>
> Thank you very much for this interesting question. We subsequently clarify our experimental setup and our deliberate choice of baselines.
>
> We agree with the reviewer that implicit regularization from optimization dynamics may act as an effective regularizer, e.g, by using lower batch sizes or gradient clipping, and believe that Lipschitz INRs are similarly amenable to such implicit regularization. Hence, a fair comparison would be challenging, as the effect cannot be easily disentangled. Additionally, Lipschitz regularization has been compared to other regularizations, such as weight decay and Dirichlet energy regularization, in prior literature \[1\], where Lipschitz regularization yielded favourable results.
>
> Moreover, the regular training objective in INRs, i.e., a pixel-wise loss, does not guide the final function to be smooth. In fact, this entirely depends on the inductive bias of the INR architecture and does not suffice to yield smooth functions for inverse problems \[2,3\]. Thus, training for longer typically makes the model more susceptible to overfitting.
>
> Lastly, we view Lipschitz constraints as complementary to other forms of regularization. In this context, the scope of this paper is not to benchmark different forms of regularization, but to develop a framework for enabling the application of Lipschitz INRs in a wide range of tasks. Hence, we deliberately focus on developing the necessary strategies for (1) estimating a reasonable Lipschitz budget and (2) distributing it to network components. These two points remain entirely absent in INR and other Lipschitz literature, and we aim to address both as key contributions of our manuscript.
>
> \[1\] Liu HT, Williams F, Jacobson A, Fidler S, Litany O. Learning smooth neural functions via lipschitz regularization.SIGGRAPH 2022\.
> \[2\] Kim N, Fridovich-Keil S. Grids Often Outperform Implicit Neural Representations. arXiv preprint arXiv:2506.11139. 2025 Jun 10\.
> \[3\] Ramasinghe S, MacDonald LE, Lucey S. On the frequency-bias of coordinate-mlps. Advances in Neural Information Processing Systems. 2022 Dec 6;35:796-809.
>
>
> >Only 5 hand-designed parametric schedules are explored with minimal justification. Given that optimal allocation appears task-dependent. The paper promises "actionable principles" but provides no mechanism for deriving allocations for new tasks.
>
> We appreciate the opportunity to clarify our method. Given the lack of prior work challenging uniform allocation, our goal is to provide a first, principled strategy using common, parametric families of allocation functions. We respectfully would like to state that these are not “hand-designed” in an ad-hoc sense, but rather are principled adaptations of standard parametric functions, commonly used in established ML practice, such as for learning rate scheduling. Moreover, by varying K\_min, we evaluated different strengths of allocations rather than just static points.
>
> Regarding your feedback on actionable principles, we have included a dedicated Section 6 with practical considerations for budget estimation and allocation. We have also ablated the proposed allocation strategies on a challenging single-image super-resolution task (SISR) for 4x upsampling, as described in Appendix B. Similar to inpainting, the proposed allocation strategies remain effective in SISR, providing significantly better quantitative and qualitative results compared to uniform allocation, as demonstrated by our statistical tests (c.f. App G4).

---

> ### Author Response · Authors · 2025-11-26
> **Response to Reviewer aw4e - Part 3**
>
> >Experiments use small samples without confidence intervals or significance testing. Claims like "significantly sharper reconstruction" rely on visual inspection despite high variance in plots, making it difficult to assess whether observed differences are meaningful.
>
> Thank you very much for bringing this to our attention. This is definitely an important point. Although we had already provided confidence intervals in our plots, our paper was missing statistical tests. We have conducted these and added them to Appendix G4. In general, these tests show that non-uniform strategies yield significantly better reconstruction quality for inpainting, and a non-uniform strategy with an appropriate K\_min achieves significantly better results than uniform allocation for registration. We have also revised the wording for the shape experiment (Exp 1 A), where we initially claimed "significantly shaper reconstruction". Since a statistical test would not be meaningful for a single sample, we have revised the wording. Thank you for bringing this to our attention.
>
> Our sample sizes for the experiments are of a similar order as in previous INRs benchmark papers \[1,2\].
>
>
>
> \[1\] Kim N, Fridovich-Keil S. Grids Often Outperform Implicit Neural Representations. arXiv preprint arXiv:2506.11139. 2025 Jun 10\.
> \[2\] Essakine A, Cheng Y, Cheng CW, Zhang L, Deng Z, Zhu L, Schönlieb CB, Aviles-Rivero AI. Where do we stand with implicit neural representations? a technical and performance survey. arXiv preprint arXiv:2411.03688. 2024 Nov 6\.
>
> We are grateful for your insightful feedback, which significantly improved our paper.
> We hope our rebuttal has clarified your questions, and we are happy to address any further concerns that remain unclear.

---

### Author Response · Authors · 2025-11-26
**General Response**

Dear Area Chair,

We sincerely appreciate the time and effort you and the reviewers have dedicated to assessing our manuscript. We are excited and encouraged by the comments characterizing our contribution as “fresh”, “theoretically well-motivated” (ane9), “conceptually elegant” (apk7), and describing our paper as “easy to follow” (m2hg) and our writing as “clear” (aw4e).

In response to the reviewers' questions and their valuable feedback, we have updated and carefully revised the manuscript. Here, we summarize the key revisions and additional experiments included in the updated manuscript (changes marked in blue). We kindly refer to our individual responses for detailed answers to specific reviewer questions.

1. **Actionable Guidelines for Practitioners:**

Reviewers (specifically m2hg and aw4e) suggested that we provide more detailed guidance on applying our Lipschitz framework to novel tasks, including the (oracle) estimation and allocation of the Lipschitz budget K.

**Action:** In the revised paper, we provide:

(i) a **dedicated section** (Section 6\) to address budget estimation and allocation in greater detail, specifically discussing strategies for novel applications, and thereby providing important insights to practitioners.
(ii) **novel experiments** (Appendix B) that demonstrate employing the proposed framework to a new application, specifically to super-resolution of natural images, demonstrating its practical value and straightforward adaptation.

2. **Lipschitz Constant Estimates of Trained Networks**

Reviewers (apk7, aw4e) asked for an empirical verification of the actual Lipschitz constants in trained networks to assess how closely these networks approach the theoretical budget.

**Action:**

We implemented the **empirical estimation** method proposed by Wang and Manchester \[1\] and updated Figs. 2, 10, 11, and 12 accordingly. Our analysis (Section 4.2) confirms that trained networks closely approach the imposed bounds. For shapes, high "Lipschitz usage" correlates with better perceptual quality (lower Chamfer Distance), particularly when using Bjoerck or SLL normalization.

\[1\] Wang R, Manchester I. Direct parameterization of lipschitz-bounded deep networks. ICML’23

3. **Motivation and Robustness for oracle estimate, learning Lipschitz budget K**:

Reviewers (apk7, m2hg) asked us to provide intuition for the proposed oracle estimate, comment on its sensitivity to noise/discretization, and the possibility of making the budget learnable.

**Action:**

(i) We **discuss** the proposed oracle estimate in the absence or presence of reference data (Section 6, Appendix C3), and detail signal-theoretic properties that enable its formulation.
(ii) We include a new **sensitivity analysis to noise** in Appendix C2, demonstrating that confidence intervals of the oracle remain robust in the presence of high noise levels, and discuss how the proposed oracle estimate effectively models discretization by design.
(iii) We **clarify** how learning the budget K subverts effective Lipschitz regularization due to contradictory optimization objectives of the INR (overfitting vs generalization).

4. **Statistical Significance, Sample Size, and Baselines:**

Reviewers (specifically aw4e, apk7, m2hg) inquired about the statistical significance of our results, the sample size, and its comparison to baselines.

**Action:**

(i) In addition to initially reported confidence intervals in plots, we now provide a **rigorous statistical analysis** of our results in Appendix G4, demonstrating the statistical significance of our allocation strategies.
(ii) We **discuss** the utility of baselines within the scope of our paper (aw4e), and additionally **provide** Jacobian regularized INRs / NODEO (m2hg, apk7) to contextualize our registration results.
(iii) We **highlight** meeting or exceeding sample sizes used in current INR benchmark papers \[1,2\].

\[1\] Essakine A, Cheng Y, Cheng CW, Zhang L, Deng Z, Zhu L, Schönlieb CB, Aviles-Rivero AI. *Where do we stand with implicit neural representations? A technical and performance survey.* TMLR
\[2\] Kim N, Fridovich-Keil S. *Grids Often Outperform Implicit Neural Representations.* NeurIPS 2025

Incorporating the reviewers' feedback has clearly strengthened our paper. Our work offers a novel, theoretically well-founded, and practically relevant Lipschitz-based framework for implicit regularization of neural networks. We are excited to share these results with the community and appreciate your consideration.

---

### Meta-Review · Area_Chair_s9g3 · 2025-12-09

**Summary:**

The reviewers gave a lot of feedback on the paper. I would classify this feedback in the following categories:
*) The reviewers praise the conceptual idea of framing Lipschitz regularization as a budget allocation problem. This idea is considered an elegant conceptual novelty. This is definitely something that readers can take away from reading the paper.
*) Clarifying questions about the details of the paper and the analysis. A lot of feedback fell into this category. The rebuttal does a good job answering many of these questions.
*) The evaluation of the practical implication. For example, there is a question of "actionable guidelines". The authors provided a partial answer to the question. Ultimately, I am not convinced that these insights are significant and I am not sure if the reviewers would be convinced. This is also linked to the next main point of using only toy examples in the evaluation. It is not clear if insights derived from toy examples are robust enough.
3) The evaluation is only performed on toy datasets. The authors argue that their choice of datasets is comparable to previous work. I would accept this as a partial answer. It is true that some previous work also focused on toy datasets, however, this is not a good enough reason to keep the standard that low.

In summary, the paper will be an interesting read for a researcher interested in regularizing INR networks, but I would believe that the broader impact, key insights, and tests at an adequate scale and for adequate applications are missing.

**Reviewer Concerns:**

I believe several of the minor concerns were addressed in the rebuttal. The fundamental concerns about the practical implications and practical impact, as well as the concerns about the evaluation, remain, at least to some extent.

**Reviewer Scores:**

I do not think the reviewers would change their scores based on the rebuttal. However, statistically, it is likely that one reviewer would have increased the score due to the ICLR dynamics of the rebuttal.

---

### Decision · Program_Chairs · 2026-01-26

Accept (Poster)